# Fine Flow Structure at the Miscible Fluids Contact Domain Boundary in the Impact Mode of Free-Falling Drop Coalescence

**Yuli D. Chashechkin *** and **Andrey Yu. Ilinykh**

Laboratory of Fluid Mechanics, Ishlinsky Institute for Problems in Mechanics RAS, 119526 Moscow, Russia; ilynykh@ipmnet.ru

* Correspondence: yulidc@gmail.com; Tel.: +7-(495)-434-0192

**Abstract:** Registration of the flow pattern and the matter distribution of a free falling liquid drop in a target fluid at rest in the impact mode of coalescence when the kinetic energy (KEn) of the drop exceeds its available surface potential energy (ASPe) was carried out by photo and video recording. We studied the evolution of the fine flow structure at the initial stage of the cavity formation. To carry out color registration, the observation field was illuminated by several matrix LED and fiber-optic sources of constant light. The planning of experiments and interpretation of the results were based on the properties of the complete solutions of the fundamental equations of a fluid mechanics system, including the transfer and conversion of energy processes. Complete solutions of the system of equations describe large-scale flow components that are waves or vortices as well as thin jets (ligaments, filaments, fibers, trickles). In experiments, the jets are accelerated by the converted available surface potential energy (ASPe) when the free surfaces of merging fluids were eliminated. The experiments were performed with the coalescence of water, solutions of alizarin ink, potassium permanganate, and copper sulfate or iron sulfate drops in deep water. In all cases, at the initial contact, the drop begins to lose its continuity and breaks up into a thin veil and jets, the velocity of which exceeds the drop contact velocity. Small droplets, the size of which grows with time, are thrown into the air from spikes at the jet tops. On the surface of the liquid, the fine jets leave colored traces that form linear and reticular structures. Part of the jets penetrating through the bottom and wall of the cavity forms an intermediate covering layer. The jets forming the inside layer are separated by interfaces of the target fluid. The processes of molecular diffusion equalize the density differences and form an intermediate layer with sharp boundaries in the target fluid. All noted structural features of the flow are also visualized when a fresh water drop isothermally spreads in the same tap water. Molecular diffusion processes gradually smooth out the fast-changing boundary of merging fluids, which at the initial stage has a complex and irregular shape. Similar flow patterns were observed in all performed experiments; however, the geometric features of the flow depend on the individual thermodynamic and kinetic parameters of the contacting fluids.

**Keywords:** drop; impact; experiment; fine structure; substance transfer

## 1. Introduction

The registration of stably reproducible components of flows in the pattern of free-falling drop coalescence with a target fluid at rest had a great impact on the development of a number of sciences such as hydrodynamics, physics in general, mathematics, biology and many others. At first, these components were ring vortices [1] and multi-tiered vortex systems [2]. With the development of scientific photography, they were supplemented by more short-lived components such as cavity, crown, sprays, and splash [3]. Publications [1–3] quickly became widespread among physicists and scientists of other specialties. The establishment of the physical nature of a number of biological patterns [4], based on sketches of the cascade of vortices formation [2], had a great influence on the development of the theory of evolution and biology in general.

With time, scientific results in the study of droplet flows were being implemented in solving applied problems of ocean acoustics [5,6], creating new technologies in oil, bio, chemical, pharmaceutical, metallurgical and other industries. At the same time, the experimental technique was developing: that is, more and more powerful controlled light sources were being created [7]. The range of illuminating wavelengths was expanding (X-ray, optical and infrared waves are now used in experiments [8,9]). Photo, film and video recording techniques were improved as well: the shooting speed of modern video cameras exceeds several million shots per second [10,11].

The number of experimental and theoretical studies began to increase rapidly, and currently, the number of annual publications on the topic of "drop impact" is approaching a thousand [12,13]. The application of high-resolution instruments allowed studying in detail the evolution of the flow components identified in the first observations [3]: in particular, the change in the shape of the cavity and crown [14,15]. The patterns of the capillary waves on their surfaces [16] was traced as well. Important features in the droplet impact flows caused by the merging of both single droplets and synchronously contacting with the target fluid pairs and even groups of drops were recognized [17].

The hypothesis of "passive admixture", which is often used in fluid mechanics in general [18], was applied interpreting observations of the impact flows of a free-falling drop as well. It allows using images of the redistribution of substance density, which is an independent physical quantity and is described by the additional equation in the complete system of fundamental equations [19–21], as an indicator of the field of a flow velocity structure. The hypothesis was used to identify compact vortices at various phases of the flow created by a fallen drop in a fluid at rest [1,2,22].

At a low contact velocity of the drop, the shape and internal structure of the area of colored liquid in the cuvette resembles a "mushroom-shaped" dusty cloud of a ground-based nuclear explosion [23–25]. In this range of parameters, the drop inflows directly into the liquid, and the cavity is formed with a delay of about 10 ms [26].

Gradually, the number of experimental, analytical, and, with the development of hard and soft technologies, numerical studies of the shape of the cavern, crown, flying spray and capillary waves began to grow rapidly. The range of parameters, in which new regimes of the flows under study were discovered, has also expanded. The conditions for the rebound of single and multiple sequentially falling small drops with a diameter $D < 1.2$ mm were experimentally determined [27]. The momentum and kinetic energy of the drop were chosen as the basic parameters. In experiments [27], single small drops did not bounce off an undisturbed surface.

Side-view photographs of the developing cavern and the surrounding crown, which is converted into a gas bubble in some flow regimes, graphs of the cavity depth as a function of time and the maximum depth as a function of kinetic energy, and estimates of the total energy components are given in [28,29]. Recording the process of a drop (colored with a thymol blue solution) merging with water allowed distinguishing new features of the flow geometry in addition to the found dependences of the maximum cavity size on the diameter and contact velocity of a drop [30]. In particular, the difference between the irregular shape of the boundary of the colored liquid region and the classical smooth cavity shell was shown. Observations of the dynamics of the process of cavity formation in a deep liquid were continued in [31,32].

The technique developed in [33] for calculating the merging of liquids, taking into account the influence of surface tension, was incorporated into a numerical model of cavity formation to describe the shape of detached gas bubbles [34]. Cavity contour calculations were also independently performed in [35]. High-frequency oscillations of detached gas cavities play an important role in the processes of generating sound packets [6,36,37], which were recorded in both air [38] and aquatic environments [39].

In the impact coalescence mode, the deepening cavity is surrounded by a crown rising above the surface of the fluid that is produced by a growing hollow cylindrical jet with a jagged outer edge [3,15]. Sequences of small droplets—sprays flying from the tops of

thin spikes at the tops of the teeth—were visualized in [40]. At the initial stage of the merger, thin layers of liquid—ejecta and lamellae—fly out from the contact of the liquids domain [41]. The expanding crown generates a group of annular capillary waves, whose patterns of propagation over relatively large distances are satisfactorily described by linear theory [42].

The synchronicity of the generation of a thin splash flying off from the cavity bottom and a submerging jet with a vortex head in the center of the cavern was noted in experiments [43]. The evolution of the flow as the drop velocity increases was traced in [44]. Calculations of cavity geometry, regime map, and comparison results with observations of pigmented droplet merging were performed in [45,46].

The high-speed video cameras of a new generation [10] allowed tracing the fine details of the flow pattern during the initial contact of a drop with a stationary target fluid. The fast evolution of the air gap shape between the merging drop and the deforming surface of the liquid, the process of formation of isolated gas bubbles, and their sequences was traced. Illumination of the liquid coalescence domain was carried out by electromagnetic waves in the optical [47,48] and X-ray ranges [49]. Depending on the size, shape, and dynamic state of the falling drop, a "necklace" of gas bubbles [47], a single central bubble [36,46], or a complex system of bubbles, the parameters of which depend on the size, speed, and shape of the drop, can be formed in the merging domain [9]. The generation of an initial (impact) high-frequency sound packet is associated with oscillations of small gas bubbles [37,39].

Estimates of the influence of viscosity on the formation of a cavern, as well as the possibility of using experimental results in describing meteorite craters on icy planets, are contained in [50].

A number of studies have studied the influence of the depth of the target fluid on the general structure and dynamics of impact flows. The formation of a cavity in a deep fluid was traced in [32]. Comparisons of the observed pattern of distribution of a tinted drop with calculations of the shape of the free surface allowed identifying different modes of formation of a "thin" and "thick" splash and determining the critical conditions for the restructuring of the flow depending on the values of the Weber and Ohnesorge numbers [8,51]. Numerical simulations have also been used to study the spreading behavior of a composite droplet [52]. A review article [53] is devoted to the analysis of the processes of a drop spreading in a thin layer of liquid and the determination of heat transfer parameters.

Calculations of the axisymmetric flow pattern, taking into account the contribution of the ASPE transformation processes during the elimination of the free surface during the merging of liquids and the formation of a new one with the flow evolution, were carried out in [54,55], following [28].

In most experiments, the flow pattern was studied when a drop falls normally onto a smooth surface of a liquid at rest. However, a number of works present the results of experimental studies of the influence of the shape of the perturbed surface, the type and dynamics of the flow on the pattern of merging of a freely falling drop. Photographs of the merging of a drop ejected along an inclined trajectory with a horizontal surface of a quiescent fluid and a classification of flow regimes are presented in [56]. The pictures of the collision of a drop with a diameter of $2.5 < D < 4.1$ mm, flying in a wind tunnel along an inclined trajectory at a speed of $U = 6.7$ m/s, with a liquid in a cuvette at its bottom, was recorded in the side and frontal projections [57]. The distortion of the cavity shape and the elongation of the spikes on its edge were traced.

The dependences of the shape of the cavity, the inclination of the splash and the direction of the preferential emission of the sprays on the position of the point of primary contact on the contour of the liquid surface deformed by a solitary wave are given in [58]. Calculations of the shape of the wave surface of a liquid by a deformed falling drop were performed in [59]. The influence of the composition of the gaseous medium on the formation of a cavity and a cloud of spray from a drop falling into a thin layer of liquid at the end of a rotating disk was studied in [60,61].

Systematic experiments have allowed identifying two modes of merging of a freely falling drop with a target fluid: intrusive, when the liquid of the drop initially smoothly inflows into the thickness of the target fluid, and impact, in which a cavity begins to form from the moment of initial contact. The condition for restructuring the flow pattern is determined by the ratio of the components of the total energy of the falling drop: kinetic energy KEn and ASPE $R_{En} = En_k/En_\sigma$. Here, $En_k = MU^2/2$ is the kinetic energy, and $En_\sigma = \sigma_d^a S_\sigma$ is the ASPE of a droplet with a diameter $D$, surface area $S_\sigma$, and mass $M$ falling at a speed $U$. Here, the coefficient of surface tension at the interface of the liquid droplet with the air environment is $\sigma_d^a$. The intrusive mode is observed at $R_{En} < 1$ [62].

In the impact mode, when the drop kinetic energy exceeds ASPE and $R_{En} > 1$, thin jets are formed in the domain where a drop merges with a thin layer of liquid on a solid substrate [63]. Thin jets were also observed in a deep fluid. The colored fibrous traces of fine jets form linear and reticular meshes the surface of the cavity and crown [64].

The formation of thin jets into the free-falling droplet spreading pattern is explained by the intrinsic features of the inhomogeneous distribution of internal energy in fluids with a free surface [65] and the properties of complete solutions of the fundamental equations system [19–21] for inhomogeneous fluids.

In today's description, one of the types of thermodynamic potentials, namely, free enthalpy—the Gibbs potential, is chosen as the main physical quantity of a fluid medium (liquid, gas, plasma) [66]. Coefficients of the Gibbs potential differential form $dG = sdT + VdP + S_\sigma d\sigma + \mu_i dS_i$ determine the traditional thermodynamic physical quantities of an inhomogeneous droplet fluid with a free surface area $S_\sigma$. Here, $V$ is the specific volume, $P(t, x_1, x_2, x_3)$ is pressure, $T(t, x_1, x_2, x_3)$ is temperature, $S_i(x_1, x_2, x_3)$ is the concentration of dissolved substances or suspended particles, $\mu_i$ is chemical potential and $\rho(t, x_1, x_2, x_3) = 1/V$ is density. All of the mentioned quantities have a clear physical meaning and are available for observation with an error estimation.

In modern fluid mechanics, some parameters of the medium, such as density and pressure, are considered to be quantities of a mixed, namely, mechanical and thermodynamic nature. Empirical relations between thermodynamic quantities constitute equations of state [66–68], among which the main one is the dependence of density $\rho = \rho(P, T, S_i)$ on pressure $P$, temperature $T$, and the concentration of dissolved substances and suspended particles $S_i$. The thermodynamic state of the medium is characterized by the spatial distributions of Gibbs potential and its derivatives, which include the density, temperature, salinity and pressure, which change over time.

Along with thermodynamic quantities, the medium is also characterized by kinetic coefficients that determine the molecular transfer of momentum (dynamic $\mu$ and kinematic viscosity $\nu = \mu/\rho$), heat (thermal or temperature conductivity coefficient $\kappa_T$) and substance (diffusion coefficient $\kappa_S$) as well as the transfer parameters of other physical quantities (velocity of sound, index of refraction and others). In weakly dissipative media, which include a large number of fluids and gases found in natural and industrial conditions (in particular, water, aqueous solutions of salts or air), kinetic coefficients have small values.

Complete solutions of the system of fundamental equations [19–21] for weakly dissipative media containing small coefficients at higher derivatives are searched by the theory of singular perturbations [69]. The calculations have shown that the complete solutions of the system of fluid mechanics equations contain functions of two classes—regular in a small parameter and singularly perturbed [70]. In periodic flows, regular solutions describe waves, and singular solutions describe the accompanying components that are fibers and highly gradient interfaces forming the fine structure of the medium. The complete dispersion relations for surface waves and ligaments in a viscous stratified liquid are given in [71]. The analysis of the intrinsic scales of singular solutions complementing inertial, internal, acoustic and hybrid waves in the fluid thickness is carried out in [72].

In the experiment, the fine structure of droplet flows on the surface and in the thickness of the target fluid was previously visualized in [64]. The fine components of flows (fibers or jets, trickles) are formed as a result of the internal or available potential surface energy

conversion into other forms. They are described by singular solutions of the fundamental equations system [71,72].

The theory and methodology of studying fluid flows was created within the framework of the "continuous medium" concept, all the physical properties of which as well as parameter of flows were described by continuous functions. This approach does not agree with the actually observed discrete atomic–molecular and finer nuclear structure of matter. Atoms and molecules forming real liquids or gases are combined into various associates of physical and chemical nature. Various structure combinations are registered in the experiments such as complexes, clathrates, clusters, and voids with individual atoms, with physical and chemical bonds [73,74]. The observations and calculations show that the internal energy and one of the forms of its representation, namely the Gibbs potential, are distributed non-uniformly in the fluid.

The linear scales of the associates are in the $10^{-8} < \delta < 10^{-6}$ cm range, where the lifetime of single elements is $10^{-12} < \tau_e < 10$ s. Each of the elements of the fluid structure is characterized by its own internal energy, which accumulates during its formation and is converted into other forms during the rearrangement of the structure and the elimination of the elements' boundaries, which ensures the fluidity of liquids [66,70]. The available surface potential energy at the liquid boundary with air or vacuum (ASPE [65]) is the most important. It is concentrated in a near-surface layer with a thickness of $\delta_c \sim 10^{-6}$ cm in droplet and target fluids. The accumulation of internal energy occurs rather slowly with characteristic free surface formation in macroscopic flows for times of the order of $10^{-3} < \tau < 10^{-2}$ s and more.

By methods of optical and X-ray reflectometry and atomic force microscopy, it was found out that the density, the dielectric permittivity, and the dipole moment in the fluid thickness and in a structurally distinguished near-surface layer with a thickness of the size order of a molecular cluster ($\delta_\sigma \sim 10^{-6}$ cm) differ noticeably [75,76]. A more complex distribution of free enthalpy could be observed in a thin surface layer with a thickness of the order of molecular size $\delta_c \sim 10^{-8}$ cm, where supramolecular structures are expressed even more clearly. In a surface layer with a thickness $\delta_b \sim 10^{-8}$ cm, the self-ionization of matter can occur due to the anisotropy of atomic–molecular interactions.

When the free surfaces are eliminated during the coalescence of liquids, the ASPE quickly transforms into other forms (at a coalescence velocity of about 1 m/s per time $10^{-10} < \tau < 10^{-8}$ s). The processes of ASPE conversion form ligaments (fast thin jets [70]), colored traces of which are observed when a free-falling drop merged with a fluid at rest in the impact mode at $En_k > En_\sigma$ [62].

The inclusion of the Gibbs potential in the description of liquids and gases flows allows taking into account the action of four energy transfer mechanisms: with currents at the velocity of $U$, with the group velocity of various waves $c_g^w$, the dissipative–diffusion one with coefficients $\nu$, $\kappa_T$, $\kappa_S$ and the internal energy conversion effects. The release and accumulation of the internal energy processes includes, in particular, transformations of available potential surface energy $En_\sigma$ of the liquid as a whole and its single structural components, which are complexes, hydrates, clathrates, clusters, voids and other combinations of atoms and molecules of physical and chemical nature [75,76].

Thin fast jets and longitudinal oscillations of the expanding circular boundary of the drop coalescence domain with a thin fluid layer on the surface of a slide were visualized in [11]. In the experiments, it was observed that the jets accelerated when they crossed the contact domain boundary of merging liquids in a cavity, which was formed in a deep liquid [77]. The evolution of the pattern of thin jets penetrating the bottom of a growing cavity at the initial stage of fluids coalescence was first traced in [78]. The purpose of this study is to visualize the flow fine structure in the vicinity of the contact domain boundary of a droplet coalescing with deep water. In the impact mode, a cavity in a deep pool begins to form at the initial contact of the droplet boundary with the free surface of the target fluid at rest if $En_k > En_\sigma$: that is, the drop KEn $En_k = MU^2/2$ exceeds the ASPE $En_\sigma = \sigma_d^a S_\sigma$.

## 2. Parametrization

The experimental methodology was developed taking into account the definition of a liquid as a fluid medium, the properties of which are characterized by the distributions of the Gibbs potential, its derivatives defining thermodynamic quantities, kinetic and other physical coefficients [66,67]. The flow is defined as the combined transfer of momentum, energy, and matter, which causes the corresponding changes in the physicochemical parameters of the medium [70]. The universal classification of flow components is based on the description of complete solutions of linearized and weakly nonlinear forms of the fundamental equations systems [70,71]. They are constructed by the theory of singular perturbations, taking into account the explicit compatibility condition. The classification includes large-scale components (waves, vortices, jets) and thin ligaments, the spatial structure of which is characterized by their intrinsic scales [72,79].

The Gibbs potentials of the droplet $G_d$, the air medium $G_a$ and the target liquid $G_t$ are included as the main dimensional parameters characterizing the flows under study, considering the complex and tunable internal medium structure, which affects the dynamics and energy of droplet flows. The indices indicate the belonging of the parameter medium. The media are also characterized by density $\rho_{d,a,t}$, kinematic $\nu_{d,a,t}$ and dynamic $\mu_{d,a,t} = (\rho\nu)_{d,a,t}$ viscosities; full $\sigma_d^a$, $\sigma_t^a$ and normalized surface tension coefficients based on the density of fluids $\gamma_d^a = \sigma_d^a/\rho_d$, $\gamma_t^a = \sigma_t^a/\rho_t$; and the equivalent diameter $D$, surface area $S_\sigma$, volume $V_d$, mass $M = \rho V_d$, momentum $p_d = MU$ and velocity $U$ of the droplet at the moment of initial contact with the target liquid.

The influencing parameters include extensive kinetic energy (KEn) $En_k = MU^2/2$ and available surface potential energy (ASPe) $En_\sigma = \sigma_d^a S_\sigma$, which are formed due to the anisotropy of atomic–molecular interactions in the drop shell. The ASPE is contained in a thin near-surface layer with a thickness of the order of the molecular cluster size $\delta_\sigma \sim 10^{-6}$ cm, and its density is $W_d^\sigma = En_\sigma/V_\sigma$. The extensive KEn density is $W_d^k = En_k/V_d$. The ratio of the drop energy components $R_{En} = En_d^k/En_d^\sigma$ can be both small and large, and the ratio of their densities $R_W = W_d^k/W_d^\sigma \sim \delta_\sigma/D$ is a small value under the conditions of these experiments.

The energy parameters of the drop, falling in a gravitational field with a free-fall acceleration $g$, also include potential energy $En_p = MgD$ in scale $D$. An additional parameter is the diffusion coefficient of the pigment, used for tinting the drop, in the target liquid $\kappa_S$.

From the system of equations and physically valid boundary conditions, it follows that the basic group of linear scales, which are determined by the physical properties of media, includes the capillary–gravitational ratio $\delta_g^\gamma = \sqrt{\gamma/g}$. It belongs to the dispersion equation of short surface waves [19] as well as to the dissipative–capillary scale $\delta_\gamma^\nu = \nu^2/\gamma$. The group of linear scales depending on the droplet velocity includes viscous velocity $\delta_U^\nu = \nu/U$, capillary $\delta_U^\gamma = \gamma/U^2$ and diffusion $\delta_U^{\kappa_S} = \kappa_S/U$ scales.

Accordingly, the first part of the time scales of the task comprises only the parameters of the medium—$\tau_\gamma^\nu = \nu^3/\gamma^2$, $\tau_g^\gamma = \sqrt[4]{\gamma/g^3}$, the second one contains the drop size—$\tau_\gamma^D = \sqrt{D^3/\gamma}$, $\tau_\gamma^{\nu D} = \nu D/\gamma$, the third part embraces its contact velocity, that is kinematic $\tau_g^U = U/g$, as well as dynamic parameters determined by the drop size $\tau_U^D = D/U$ and the thickness of the shell $\tau_U^\sigma = \delta_\sigma/U$. The duration of the transferring matter and surface energy processes of the merging drop is determined $\tau_U^D = D/U$ by the drop size and the thickness of the subsurface layer $\tau_U^\sigma = \delta_\sigma/U$.

The process of converting ASPE into other forms with the elimination of the free surface of the merging liquids proceeds in a short time of the order of $\tau_U^\sigma \sim \delta_\sigma/U \sim 10^{-8}$ s for typical conditions of the experiments with free-falling droplets. Rapid processes of ASPE transformation into other forms at the circular boundary of the fluid coalescence contribute to the formation of thin jets and the generation of capillary waves in the target fluid [11,42,78].

A large number of scales of the same dimension reflect the diversity and complexity of processes occurring in a wide range of scales—from supramolecular ones of the order of

$\delta_\sigma \sim 10^{-6}$ cm in the processes of release and accumulation of ASPE to the full size ones of the flow domain.

The relations of scales of the same dimension define a set of traditional dimensionless parameters, which includes the following numbers: Reynolds Re $= D/\delta_U^\nu = UD/\nu$, Froude Fr $= En_k/En_p = U^2/gD$, Bond Bo $= D^2/\left(\delta_g^\gamma\right)^2 = gD^2/\gamma$, Ohnesorge Oh $= \sqrt{\delta_\gamma^\nu/D} = \nu/\sqrt{\gamma D}$, Weber We $= D/\delta_U^\gamma = DU^2/\gamma$, and Schmidt Sc $= \nu/\kappa_S$. The relations of the energy components form additional dimensionless combinations $R_{En}^{k,\sigma} = En_k/En_\sigma$ and $R_W = W_d^k/W_d^\sigma \sim \delta_\sigma/D << 1$.

The intrinsic scales of the task determine the requirements for choosing the size of the observation area, the spatial–temporal resolution of the instruments, and the duration of recording the flow pattern. Dimensionless relations allow us to assess the relative contribution of processes of different nature to the overall flow pattern and compare the conditions of independent experiments.

## 3. Experimental Setup

The experiments were carried out on a modified stand for studying the fine structure of fast processes (FSPs), which is part of the Unique Research Facility "HPC IPMech RAS" [80]. The installation, which is presented in Figure 1, included a cuvette with a target fluid 1, time delay unit 2, fiber-optic light source with fiber light guides 3, a recording photo or video camera 4, matrix LED sources 5, monitor and a computer for controlling the experiment and for data collecting 6, drop fluid discharge valve 7, drop generating dispenser with normally cut capillary 8, and a photodetector 9 that marks the pass of a drop intersecting the light beam. White dashed lines are the vertical trajectory of the drop and the inclined line of sight of the optical system.

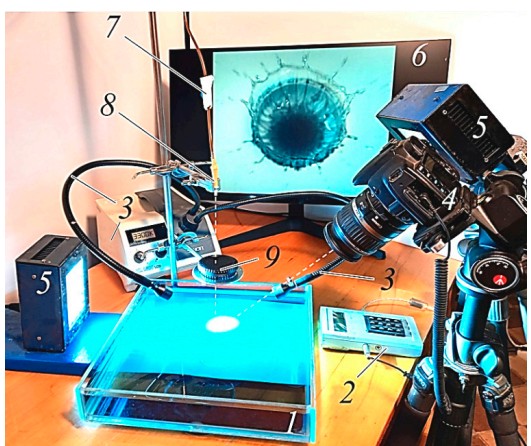

**Figure 1.** Experimental setup: 1—cuvette with a target fluid, 2—time delay unit, 3—fiber-optic light source with fiber light guides, 4—photo or video camera, 5—matrix LED sources, 6—monitor and a computer, 7—discharge valve, 8—dispenser, 9—photodetector.

The drop formed in a dispenser *8* with a replaceable capillary of a diameter $0.8 < d_o < 4$ mm with a flat cross-section broke off under the gravity and fell on the surface of partially degassed tap water in a target cuvette *1* of the size $10 \times 10 \times 7$ or $30 \times 30 \times 5$ cm$^3$. The pattern of coalescence of single water drops, aqueous solutions of alizarin ink or metal salts (potassium permanganate $KaMnO_4$, copper $CuSO_4$ and iron $FeSO_4$ sulfates) was studied. The target fluid, which was partially tinted with a drop pigment, was renewed after each experiment.

The registration system included an Optronics CR $300 \times 2$ video camera or a Canon EOS 350D camera, the sight line position of which was selected in favor of the greatest clarity of the boundaries of the recorded pattern components. Two directions of the sight line were mainly implemented: at an angle to the horizontal $\vartheta = 0°$ when registering the

flow pattern in the vertical plane or $\vartheta = 65°–70°$ when observing the free surface (the distance from the lens to the center of the flows was from 12 to 40 cm, the pixel sizes in the experiments ranged from 10 to 50 microns). The shutter speed was chosen to be minimal for a given level of spatial resolution, the size of the recorded area and the required illumination.

The fluids merging domain was illuminated by two Optronis MultiLED matrix constant light sources with a luminous flux of 7700 lm and also two fiber-optic sources of the Schott KL2500LCD (the power was 250 W, the luminous flux was 1300 lm). Special attention was paid to the organization of lighting in order to exclude the shading of the examined components of the flow or the effects of total internal reflection.

Traditionally, the technique of "back-light" illumination with a bright continuous [81] or pulsed [82] light source with an arc, gas-discharge or stroboscopic lamp is used in the droplet flows observation experiments. Recently, LED [44], fiber-optic [62] and laser [83] sources have become widespread. The emitted light is directly projected on to the studied flow domain, and sometimes, it passes through a semitransparent screen (for example, tracing paper [47]) for more uniform illumination of the field of view. To increase the spatial resolution of the flow pattern components in the fluid thickness, a light laser knife and shadowgraphy were used as well [84].

In these experiments, combined lighting, including multi-element LED and fiber-optic sources, was used to ensure the contrast of the droplet residue image, the continuously changing cavity shape and the distribution of the droplet matter in the liquid. The location of the light sources was chosen based on the maximum contrast of the residue image at the bottom of a continuously growing cavity. The simultaneous rapid change in the location and shape of the flow components led to some defocusing of the image, which made it difficult to conduct further analysis. The stability of the vertical position of the contact line moving along the surface of a thin fluid layer on the glass slide contributed to the improved clarity of the video shots [11].

When the detached drop crossed the light ray, the signal of the photodetector launched the registration system with the selected time delay. The droplet velocity was estimated by its displacements in the last shots before the contact with the target liquid and the duration of the signal delay from the photodetector to start the flow pattern recording. The image was scaled according to photos of the marker photos with size from 1 to 10 mm. Photometry and data processing were carried out in the Matlab and Kompas envelope (CAD).

## 4. Main Results

In each series of experiments, the individual adjustment of several sources was used instead of the traditional "back-light" illumination for the flow pattern lighting. Simultaneous visualization of the shape of a rapidly deforming free surface during the formation of a cavity, a crown, a veil on its edge, as well as the registration of the drop substance position was organized. The lighting technique allowed us to carry out both color and black-and-white registration of the flow pattern. To control the universality of the proposed flow structural features classification, the spreading of uniformly colored drops of dilute solutions of alizarin ink, potassium permanganate, copper and iron sulfates, as well as tap water was studied. In all cases, the image geometry of both the moving free surface and the drop matter distribution in a target fluid was recorded. While the overall fine flow structure was preserved, the individual properties of small elements turned out to depend on the composition of the media.

### 4.1. Drop Spreading of Alizarin Ink Solution in Water

The evolution of the fine structure of the matter transfer pattern during the free-falling drop coalescence of a dilute 1:200 solution of blue alizarin ink is illustrated by the video shots shown in Figure 2. The line of sight in these experiments is inclined at an angle $\vartheta = 65º$ to the horizon. The shooting speed is 4000 fps, the shutter speed is 1/5000 s, and the spatial resolution is 30 microns/pixel (μm/pix).

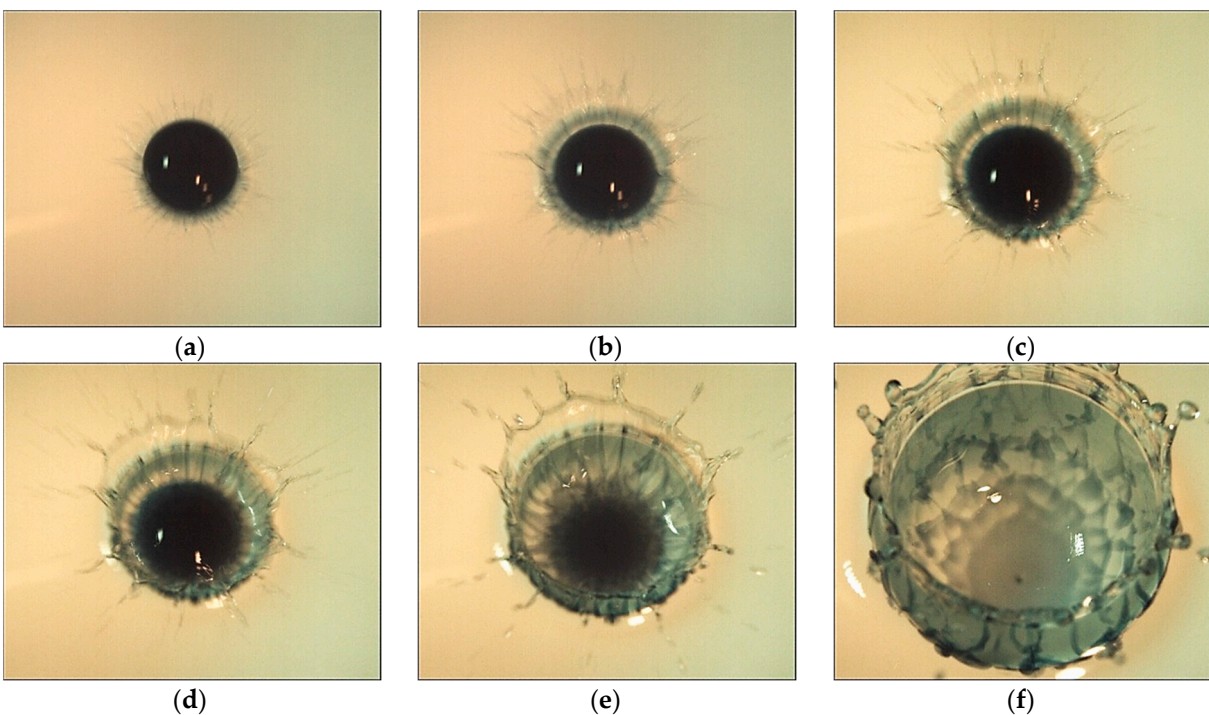

**Figure 2.** Evolution of the drop fluid distribution pattern (an aqueous ink solution diluted in 1:200 ratio) in water ($\rho_d = 1$ g/cm$^3$, $\sigma_d^a = 73$ g/s$^2$, $\nu = 0.01$ cm$^2$/s, $D = 4.3$ mm, $U = 3.1$ m/s, $En_\sigma = 4$ µJ, $En_k = 200$ µJ, Re = 13,300, Fr = 230, We = 570, Bo = 2.5, Oh = 0.0018, R$_{En}$ = $E_k/E_\sigma$ = 48, R$_W$ = $1.66 \cdot 10^{-3}$). (**a**)—$t = 0.2$ ms, (**b**)—$t = 0.45$ ms, (**c**)—$t = 0.7$ ms, (**d**)—$t = 1$ ms, (**e**)—$t = 1.7$ ms, (**f**)—$t = 6$ ms.

The primary contact of the drop is accompanied by the release of a veil (a gray strip $\Delta r_v = 0.45$ mm wide along the main diameter $d_{cr} = 5.6$ mm), which is pierced by thin spikes. A sequence of drops is thrown from the tops of the spikes (Figure 2, $t = 0.2$ ms). The deviation of the pattern from strict symmetry is due to the inclination of the line of sight and some asymmetry of the merging drop shape.

The lengths of separate jets, which are elongated traces of thrown droplets, is four times the width of the strip. In general, the trace pattern radially diverges, but some colored jets are located at an angle to the local radius vector. As it follows from the analysis of variations in the illumination distribution pattern along an arc $\Delta r_l = 0.2$ mm away from the contact line, the thickness of the droplet traces is $d_s \sim 50$ microns, the length reaches $l_s \sim 1.8$ mm, and the inclination angle from the direction of the local radius vector lies in the range $0 < \varphi < 35°$. The dispersion of the angular positions of the spray trajectories can be caused by variations in the angle between the shells of the veil edge in the vicinity of the spike from which the small sprays are thrown out.

The general asymmetry of the flow pattern is because of the difference in the shape of the incoming drop and the spherical one due to Rayleigh oscillations and capillary waves traveling along its surface [85]. The general gray background of the veil and darker lines, which are traces of the thrown liquid layer and single ligaments (thin jets that arise at the contact line of the drop with the target liquid), indicate the complexity of the emerging flow pattern. It expresses both two-dimensional continuous veil and fast spikes that are three-dimensional components.

The ratios of the sizes (length and diameter) of individual strokes, which are the extensions of spikes (blurred images of thrown droplets—their velocity is $u_s = l_s/\Delta t$, $l_s \sim 1.8 \div 2$ mm—length of droplet traces, $\Delta t = 0.25$ ms—exposition time, $u_s \sim 7 \div 8$ m/s), show that their velocity noticeably exceeds the drop velocity [86]. The rapid flow of liquid in the jets is provided by the additional ASPE released when the free surface of the merging

liquids is eliminated. This energy is transformed into other forms, including the energy of mechanical motion.

During the drop coalescence, the flow pattern, in which one can distinguish the veil in the upper left part and the crown, becomes more complicated (Figure 2b, $t = 0.45$ ms). The visible edge of the tinted domain (the trace of the drop residue) loses its smoothness. Colored fibers come out of individual protrusions, which spread at the cavity bottom, go through the crown wall and the veil, and form spikes on its outer border. The continuity of the jet traces (linear elements of the flow geometry) indicates the structural stability of the developing flow. Jets are formed at the initial coalescence phase and persist for a long time.

The radial inhomogeneity of the image brightness allows us to confidently identify at $t = 0.45$ ms the individual flow components. They include the drop residue with a diameter of $d_r = 4.74$ mm and the cavity bottom. A dark circular line is the inner boundary of the crown wall; the circular boundary line is the crown outer edge, which is continuing by veil. The outer edge of the veil is tightening and thickening; individual fibers on its surface and their extensions are spikes coming out from the tops of the teeth. Strokes, which proceed the spikes, are traces of fast flying sprays.

The jet wakes (dark fibers) are adjacent to the protrusions at the visible contact boundary of the drop residue and the deepening cavity. The slope of the cavity bottom explains some clarity loss of the image of the crown moving boundary and the drop residue spreading along the cavern bottom. However, here, the extension of jet traces connecting the spikes on the veil edge with the boundary of the drop residue is well traced.

As the flow evolves, the diameter of the central spot slowly grows (the drop remnant spreads along the cavity bottom). At the same time, the adhesion of the colored fibers to the boundary of the merging liquids is more and more clearly expressed (Figure 2c, $t = 0.7$ ms).

With time, the diameter of the droplet residue (the central undisturbed spot) decreases somewhat due to the formation of new tiers at its edge of a discrete distribution pattern of the droplet matter (Figure 2d, 1 ms). The boundary between the drop residue and the cavity bottom becomes more and more indented and thickened. There are separate protrusions on it that are analogs of spikes on the veil edge in the air. Over time, the diameters of the protrusions grow, and the contrast decreases.

The elements of the fibrous reticular structure are traced inside the tinted central spot with a diameter of $d_r = 5.45$ mm at $t = 1.7$ ms. Gradually, the linear structure on the outer cavity boundary is replaced by a reticular one; at $t = 6$ ms, triangular cells appear on the crown walls (Figure 2f). Five tiers of cells are allocated inside the cavity.

The photometry results of the relative illumination $I(l_\varphi)$ in the flow pattern presented in Figure 2e at $t = 1.7$ ms and the dependence of the relative energy spectrum $S(\lambda)$ on the scale $\lambda$ are shown in Figure 3. The illumination values $I(l_\varphi)$ are determined in a circular spot with a diameter $d_p = 30$ μm, which moves a distance $l_\varphi$ along the arcs of circles with a radius $r_\varphi = 3.15$ and 2.6 mm.

In the middle of the cavity sidewall at $r_\varphi = 3.15$ mm, where linear structures in the distribution of the drop matter are distinguished, the peaks on the spectrum correspond to scales $\lambda = 0.42, 0.47, 0.54, 0.63$ and 0.8 mm. In the transition zone between the drop residue and the cavity bottom at $r_\varphi = 2.6$ mm, where reticular structures with triangular cells are traced, the peaks at scales $\lambda = 0.39, 0.49, 0.6$, and 0.71 mm are distinguished in the spectrum. The smallest scales characterize the thickness of single fibers.

As can be seen in Figure 2, and a number of subsequent ones, the color image of the drop coalescence expresses a number of structural components identified in [67,77], which were not previously studied systematically. For convenience of further reading, photographs of the flow are given in Figure 4 with designations of the components under discussion and an indication of their sizes.

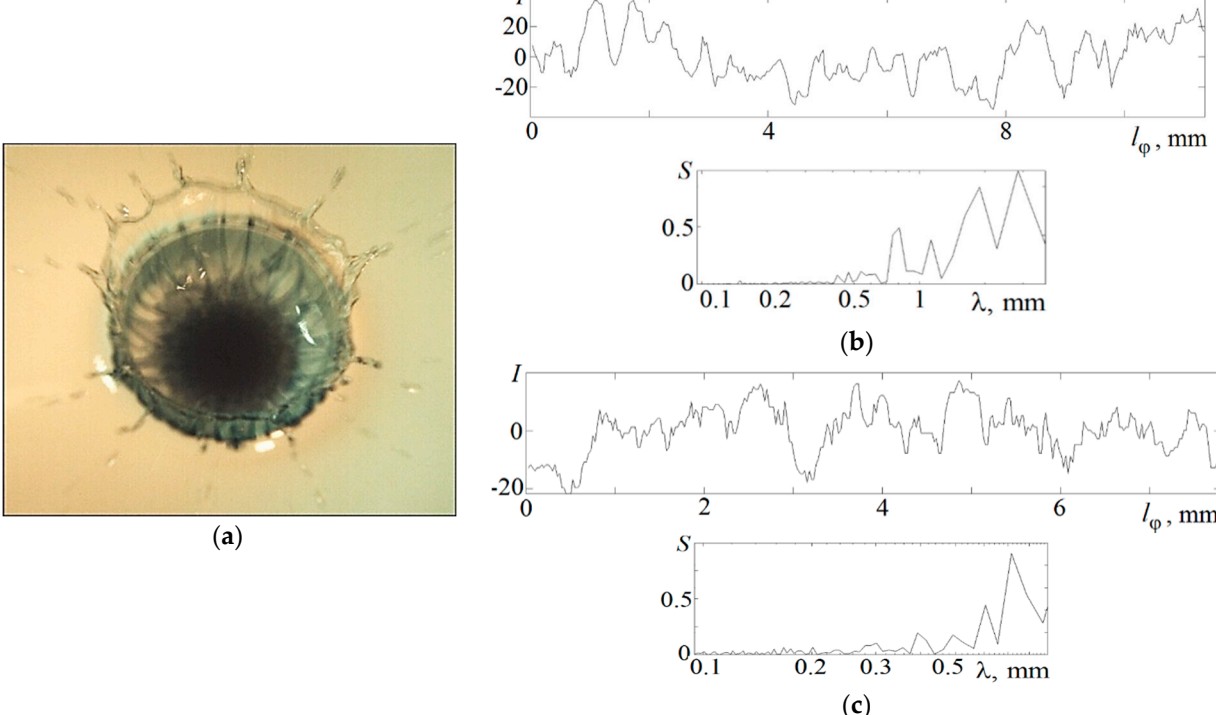

**Figure 3.** The fine flow structure: (**a**) shot at $t = 1.7$ ms; (**b**,**c**) relative illumination distributions $I(l_\varphi)$ along the $r_\varphi = 3.15$ and 2.6 mm circumference arcs and their spectra $S(\lambda)$.

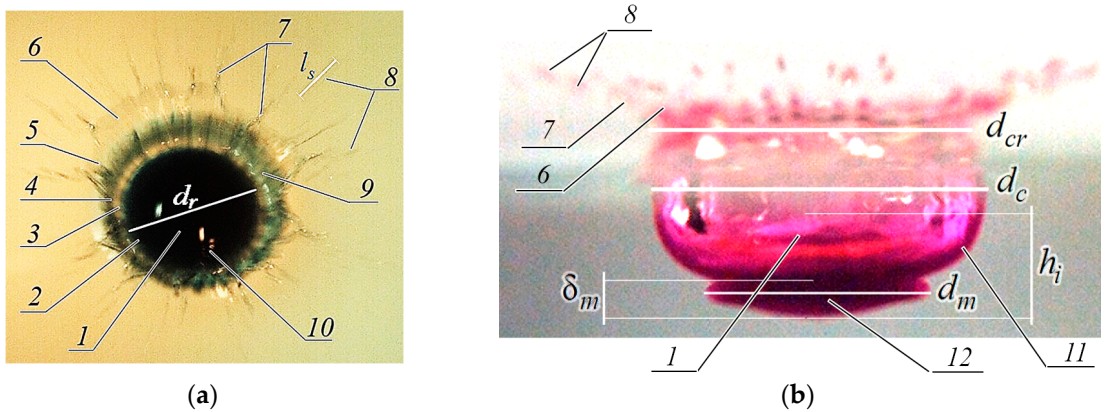

**Figure 4.** Structural components of the flow pattern in the impact mode of droplet coalescence: (**a**,**b**) frontal and side view (1 is the remainder of a coalescing drop with diameter $d_r$; 2 is the boundary of the fluid merging domain; 3 is the bottom of cavity; 4 is the area of the cavity to the crown transition; 5 is the outer boundary of the crown edge; 6 is a veil; 7 are spikes; 8 are traces of droplets (sprays); 9 are ligaments—fibrous traces of colored jets on the walls of the cavity and crown; 10 is the system of short capillary waves on the bottom of the submerging drop, 11 is densely colored layer at the cavity, 12 is intermediate layer. (**a**)—$t = 0.7$ ms. (**b**)—$t = 2.25$ ms.

In the frontal image of the drop spreading pattern in Figure 4a, the following structural components are indicated: the remainder of a coalescing drop *1* with diameter $d_r$ at the bottom of the cavity; the boundary of the fluid merging domain of complex shape *2*; the bottom of cavity *3*; the area of transition of the cavity to the crown *4*; the outer boundary of the crown edge *5*; *6* is a veil; spikes are in *7*; the long stretched images of rapidly flying droplets (sprays) *8*; fibrous traces of colored jets on the walls of the cavity and crown—ligaments *9*; a system of short capillary waves on the bottom of the submerging drop *10*. Wakes of jets are continuous fibers colored by the substance of the drop, which are stretched from

the boundary of the area where liquids merge to the edge of the crown and the veil. On the edge of the veil, continuations of the jets form spikes *7*, from the tops of which small, partially colored droplets *8* fly out sequentially. The lengths of several strokes (traces of small droplets) *I–III*, highlighted on the right side of the figure, are equal to $l_s = 1.77$, 1.87, and 1.82 mm, respectively. Some spikes are turned into the flow center. The sprays flying from their tops form capillary waves on the back surface of the submerging drop *10*. Traces of the spray collision ы with the backside of the submerging drop were also visualized in [87].

In the lateral projection, only part of the flow structure components designed with the same numbers is presented. Sprays from different spikes and remnants of spike fly synchronously. Droplets on the tops on the edge of the crown with a diameter of $d_{cr}$ are visible. Its outer boundary is marked by 5 in Figure 4a

Under synchronously flying sprays from different spikes and remnants of spikes, droplets on the tops are visible on the edge of the crown with a diameter of $d_{cr}$ (its outer boundary is marked by *5* in Figure 4a). In the underwater part of the flow, there is a cavity with a diameter of $d_c$ (corresponding to 4 in Figure 4a).

In the bottom of the cavity, an "intermediate layer" 12 with a diameter $d_m$ and a height $\delta_m$ is adjacent. A dark color and a clear boundary structurally distinguish the "intermediate layer". The symbol $h_i$ indicates the height of the densely colored layer in the lower part of flow 11, which includes the remnants of the drop at the cavity bottom and the adjacent intermediate layer.

The evolution of flow parameters is illustrated by the graphs presented in Figure 5. In the phase of active cavity growth, the outer diameter of the crown, represented by curve *1* in Figure 5a, grows monotonically and is approximated by the exponential function $d_{cr}(t) = 4.5t^{0.5} + 4$.

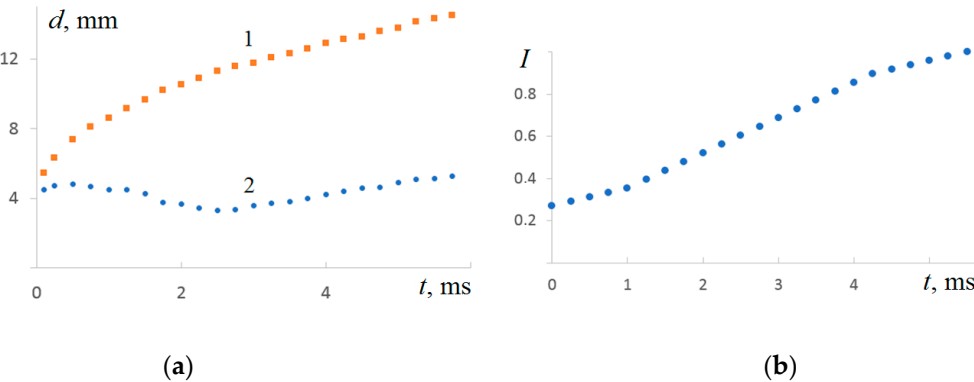

(**a**)  (**b**)

**Figure 5.** Evolution of parameters of the ink pigmented drop coalescing with water: (**a**) curves *1, 2* are diameters of the crown $d_{cr}(t)$ and the central colored spot $d_r(t)$; (**b**) changes in illumination in the center of the flow inside a circle with a diameter $d_p = 0.8$ mm.

The size of the central densely colored spot $d_r$ changes non-monotonically with time. At first, it increases somewhat; then, in the phase of merging of the bottom part of the drop, it decreases somewhat and begins to increase again at $t > 2.5$ ms. The observed increase in the size of the spot is explained by the spreading of the previously formed intermediate layer, colored by the pigment of the drop, along the bottom of the growing cavity.

In this case, due to the dilution of the pigment and the spreading of the colored liquid, the illumination in the center of the cavity $I(t)$ gradually increases (Figure 5b). Three sections are highlighted in the dependence graph $I(t)$: slow growth at $t < 1$ ms, accelerated at $1.0 < t < 4$ ms and again slow at $t > 4$ ms.

The change in the rate of evolution of illumination is explained by the complexity of competing diffusion and hydrodynamic processes occurring on the cavity wall and in the intermediate layer under its bottom. With further evolution, the protrusions on the cavity

sidewall are transformed into short jets, which, when the cavity collapses, form elongated colored loops in neutral and chemically reacting contacting media [88].

The pattern features of a drop spreading, colored with blue alizarin ink, in the fluid thickness are illustrated by frames from the video of the side view flow pattern shown in Figure 6. Drops begin to deform the free fluid surface, creating a cavity, from the moment of the initial contact. At the same time, the drop liquid leaks in thin jets through the cavity bottom and forms an intermediate layer in which the fibers of the colored liquid are separated by interfaces of the target fluid [78]. Over time, diffusion equalizes the difference in densities of the pigmented liquid of the drop and the target fluid, evens out the pigment distribution and forms an intermediate layer adjacent to the bottom of the cavity. The formed intermediate layer of liquid of its own density, separated from the receiving liquid by a high-gradient shell, has also been previously visualized in [84,89].

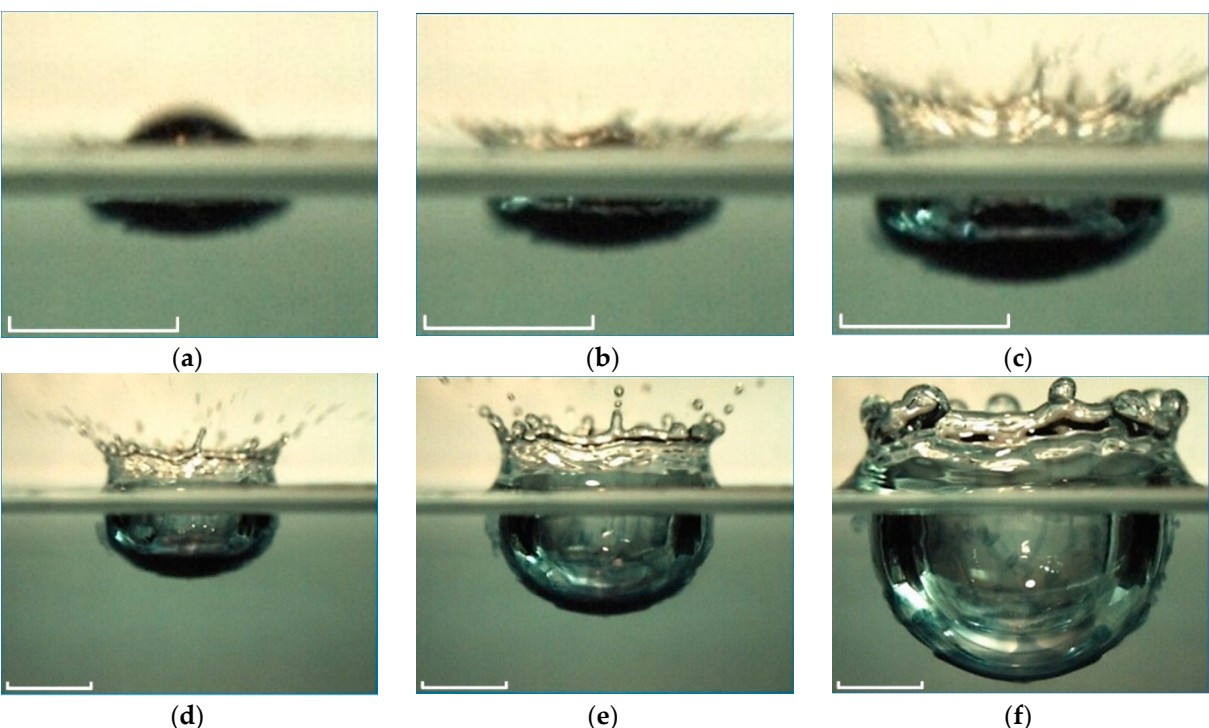

**Figure 6.** Side view of the flow pattern and the impurity distribution when a drop of ink solution is immersed in water (the experimental conditions are shown in Figure 2; the tag length is 5 mm). (a) $t = 0.75$ ms, (b) $t = 1$ ms, (c) $t = 1.75$ ms, (d) $t = 2.75$ ms, (e) $t = 5$ ms, (f) $t = 14.25$ ms.

In the first shot of the flow pattern in the lateral projection shown in Figure 6a, a dark strip with a flat bottom $d_c = 5.6$ mm in diameter, and $h_c = 0.56$ mm in height, adjacent to the free surface, visualizes the cavity partially covered with droplet liquid fibers.

Adjacent to the cavity from below is a growing intermediate layer with diameter $d_m = 3.5$ mm and height $h_m = 0.4$ mm, which is formed by fibers containing a drop matter that have passed through the cavity bottom. The fibers grow perpendicular to the cavity bottom at the initial stage of the coalescence process, when the size of the contact domain of liquids increases rapidly. In this case, the free surfaces of the contacting liquids are eliminated with the simultaneous conversion of ASPE into other forms, causing the formation of thin fast jets. The visualization of the intermediate layer fine structure at the later stages of droplet spreading was given in [84], and its thickness is measured in [89].

The fibrous structure of the flow pattern, which is presented along the entire boundary of the intermediate layer, is more clearly expressed on its lower edge in Figure 6b. In the upper part of the cavity with diameter $d_c = 6.12$ mm and depth $h_c = 0.96$ mm, an air

layer of cavity with thickness $h_i = 0.5$ mm is visible. An intermediate layer with diameter $d_m = 3.9$ mm and thickness $h_m = 0.44$ mm adjoins the cavity from below.

As the cavity grows, the intermediate layer on its lower edge becomes thinner, and its outer border is leveled (Figure 6c). Here, the main part of the droplet substance is concentrated in the lower part of the cavity in a layer $h_i = 0.6$ mm high, including an advanced intermediate layer. Examination of the enlarged image shows that the lower part of the cavity is densely covered with colored liquid droplets, and in the upper part, single colored fibers are distinguished.

Furthermore, the intermediate layer becomes much thinner and covers the cavity surface with diameter $d_c = 8.7$ mm with a thin layer (up to $h_i = 0.3$ mm thick; see Figure 6d). The cavity height with an intermediate layer is $h_c = 3.42$ mm. Here, the intermediate layer is distributed more evenly along the cavity bottom; its thickness does not exceed $h_i = 0.4$ mm. In a densely colored domain, the droplet matter is collected in the lower part of the cavity in a $\Delta h_c = 0.9$ mm high layer. At the upper edge of the intermediate layer, the surface is tinted most densely. Crests and troughs $h_w \sim 2$ mm in size are traced on the border of the tinted layer.

After the crown reaches its maximum size (the height is $h_{cr} = 3.7$ mm, the diameter is $d_{cr} = 11.3$ mm) and begins to decline, the capillary waves propagate down from its upper edge. The shape of the phase surfaces of waves replicates the changing contour of the edge crown. As the cavity deepens further, the concentration of the drop pigment becomes more and more aligned in height (Figure 6e). Gradually, the pigment flows down, the side wall of the cavity become sufficiently transparent and vertical fibers are displayed on them (Figure 6f). At the same time, the lower edge of the intermediate layer begins to lose its smoothness again; protrusions and troughs appear in it. They are created by the flow of colored fluid along the fibers. The fluid accumulates at the reticular formation nodes, penetrates through the cavity surface, and forms small vortices that are gradually transformed into fiber loops [88]. At the same time, the crown edge is rounded, and capillary waves cover its entire surface.

The release of droplets from the tops of the spikes on the crown surface ceases over time; only their rounded remnants persist (Figure 6f, $t = 14.25$ ms). The inner surface of the cavity is covered with individual fibers and spots containing droplet matter. The intermediate layer on the outer surface of the cavity also loses its homogeneity; individual brightly colored domains appear in it.

The study of color flow patterns allows distinguishing between the processes of deformation of the liquid surface forming a cavity and a crown and the processes of droplet matter transfer. The fibers containing the droplet matter leak through the deformed boundary of the cavity and form a finely structured layer under the cavity bottom in the thickness of the target fluid. Gradually, diffusion aligns out the heterogeneity of the droplet pigment distribution, and a uniformly colored intermediate layer is formed under the cavity bottom. In the frontal projection, it corresponds to a more densely colored spot in the flow center. Over time, the thickness and size of the spot, as well as its shape, undergo significant changes due to the size and shape alteration of the cavity, the fluid flow in thin jets forming linear and reticular structures on the cavity inner surface and in filament loops under it.

### 4.2. Drop Spreading of Potassium Permanganate Solution in Water

The video shots of the drop coalescence of dilute potassium permanganate solution with water are presented in Figure 7 (the inclination of the line of sight is $\vartheta = 65°$ to the horizon, and the shooting speed is 4000 fps). When the chemical composition of the merging liquids changes, the general flow structure is preserved, but the parameters of the thin components change noticeably. The outer ring of the flow at the initial droplet contact consists of individual spikes $\delta_s \sim 0.1$ mm thick and $l_s < 2$ mm long, which are partially tinted with pigment droplets (Figure 7a, $t = 0.25$ ms).

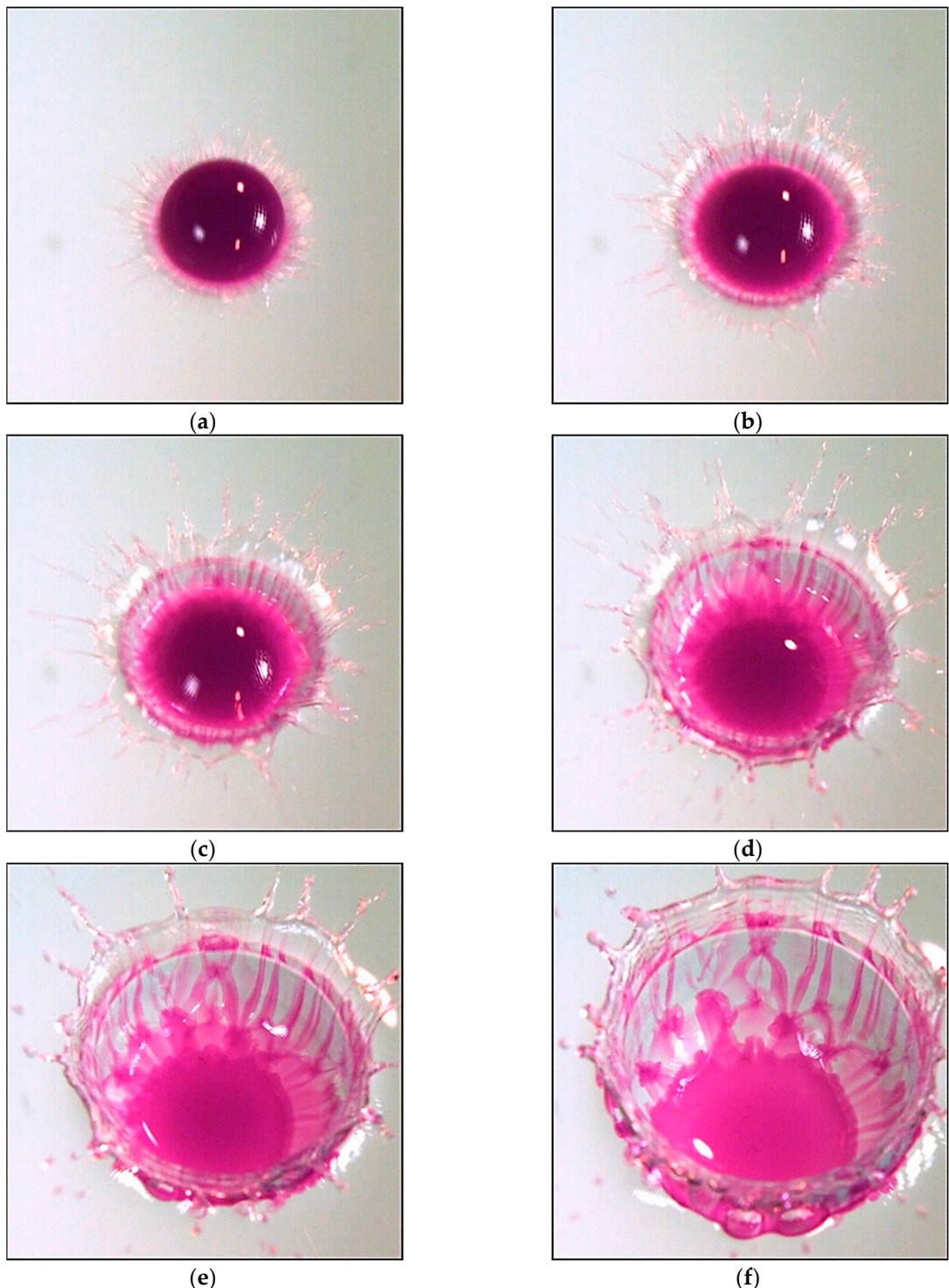

**Figure 7.** The flow pattern evolution on the water surface at the drop of aqueous potassium permanganate solution diluted in a 1:100 ratio coalescence ($\rho_d = 1$ g/cm$^3$, $\sigma_d^a = 73$ g/s$^2$, $\nu_d = 0.01$ cm$^2$/s, $D = 4.3$ mm, $U = 3.1$ m/s, $E_\sigma = 4.2$ µJ, $E_k = 200$ µJ, Re $= 13,300$, Fr $= 230$, We $= 570$, Bo $= 2.5$, Oh $= 0.0018$, R$_{En} = En_k/En_\sigma = 48$, R$_W = 1.66 \cdot 10^{-3}$). (**a**) $t = 0.25$ ms, (**b**) $t = 0.5$ ms, (**c**) $t = 0.75$ ms, (**d**) $t = 1.25$ ms, (**e**) $t = 2.25$ ms, (**f**) $t = 3.5$ ms.

The spikes are in contact with the edge of a thin, weakly colored light circular veil with a width $\delta_v = 0.57$ mm, in which thin brightly colored jets $\delta_l \sim 0.1$ mm are traced. They form a rather regular linear structure, which has been repeatedly observed in experiments

with other substances [86]. Large splashes of light allow considering the veil surface as a continuous relatively smooth surface.

The veil is adjacent to a darker wide ring $\delta_{cr} = 0.2$ mm visualizing a growing crown. A brightly colored strip with a width $\delta_{ca} = 0.18$ mm is a growing cavity in contact with a drop residue $d_r$ = 4.5 mm in diameter.

The examination of the enlarged image shows that colored fibers protrude from under the drop and continuously last in the cavity, crown, and veil. They form the core of a thin jet, which is a spike protruding from the crown edge. Thus, the flow of the spreading droplet has a predominantly radial direction.

The flow structure is preserved in the next shot at $t = 0.5$ ms. The dimensions of the above-mentioned structural components have grown: they are $\delta_s \approx 0.15$ mm and $0.5 < l_s < 2.5$ mm for the thickness and length of the spikes, $\delta_v = 0.74$ mm for the veil, $\delta_{cr} = 0.4$ mm for the crown and $\delta_{ca} = 0.36$ mm for the cavity. The diameter of the drop residue (a dark area with an uneven edge), the protrusions of which are adjacent to the colored fibers, is $d_r = 4.26$ m. The outer edge of the veil becomes more indented, the number of spikes increases (in the first frame you can count 17; in the second one, there are 19 teeth in the upper semicircle).

Over time, the clarity of the fiber pattern at the bottom of the cavity remains, which becomes more and more densely colored (Figure 7c). At the same time, the boundary between the cavity bottom and the drop residue with diameter $d_r = 4.33$ mm becomes increasingly indented. The preservation of the continuity of the colored fibers indicates the immutability of their angular position in the moving contact domain of the merging fluids.

As the crown width increases and the cavity deepens, the velocity of the colored jets decreases, the protrusions in the edge contour of the cavity deepen, and the spikes break up into sequences of individual droplets, the diameter of which grows over time (Figure 7d). A reticular pattern of individual fibers is visible under the colored drop liquid floating to the cavity bottom. The uneven boundary of the central spot shifts slightly to the flow center—$d_r = 4.6$ mm.

As the cavity deepens and the crown expands, the contrast of the fibers decreases, and their width increases (Figure 7e, $t = 2.25$ ms). A fracture in the angular position of the fibers indicates a sharp change in the inclination of the cavity walls, at the bottom of which the contours of the emerging reticular structure of the fibers are traced in a diffusely colored layer. In the drop residue, a densely colored core and a diffuse outer part are distinguished. The fibers move to the protrusions of its boundary. They are traced on the cavity walls, on remnants of the veil, spikes, and ejected sprays. The outer part of the cavity sidewall, visible in the lower part of the figure, loses its smoothness. Jets begin to bulge on it, gradually transforming into fibrous loops when the cavity collapses [88].

After a while, the contrast of the fibers is decreasing, and their width is increasing (Figure 7f). The color of the flow central domain becomes more uniform. The inhomogeneity of the outer boundary contour of the drop residue is preserved in the right part of the figure, where the fibers are visualized along the entire length. In general, the pattern of fiber distribution is being rebuilt, and its elements are being enlarged.

In the energy spectrum $S(\lambda)$ of the distribution of the relative illumination flow pattern $I(l_\varphi)$ along a circle with a radius $r_\varphi = 2.75$ mm in the upper half-ring shown in Figure 8 at $t = 1.7$ ms, the peaks at scales $\lambda = 0.2, 0.37, 0.44$, and $0.76$ mm are distinguished. The smallest scales characterize the thickness of individual fibers.

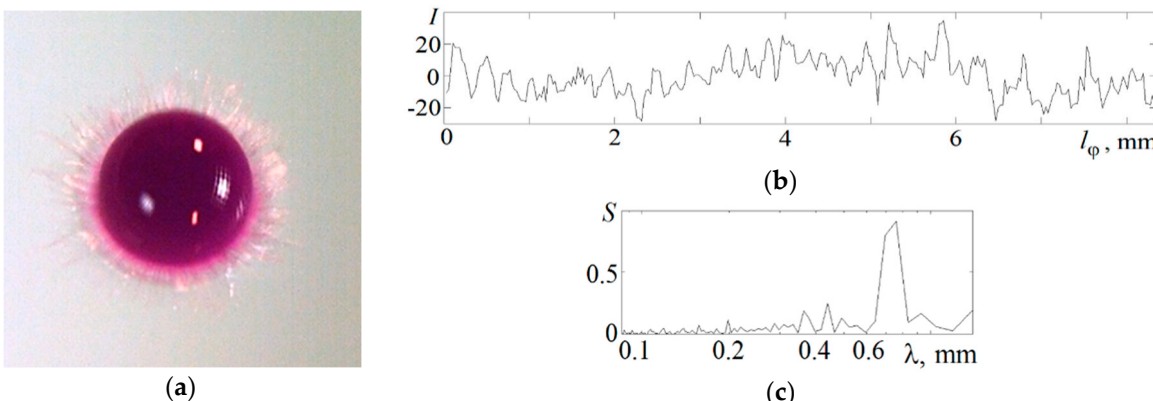

**Figure 8.** The fine structure of the flow: (**a**) the shot from Figure 6, $t = 1.7$ ms; (**b**,**c**) the distribution of relative illumination $I(l_\varphi)$ along the arc of the $r_\varphi = 2.75$ mm circle, in the middle of the side wall and in the transition zone between the drop residue and the cavity bottom, and their spectra $S(\lambda)$.

The peculiarities of the droplet matter transfer in the impact coalescence mode are explained by samples from the video film of the flow pattern in the vertical plane (Figure 9, side view image). At the initial stage of droplet coalescence, the flat bottom of the cavity is penetrated by thin fibers separated by interfaces of the target fluid. A fibrous layer of diameter $d_m = 3.44$ mm and height $\delta_m = 0.3$ mm with an uneven lower edge adjoins a flat brightly colored bottom of a growing cavity with width $d_c = 4.83$ mm and height $h_c = 0.52$ mm (Figure 9a). The cavity quickly deepens, and at $t = 0.5$ ms, a gas cavity with width $d_a = 3.3$ mm and height $h_a = 0.25$ mm begins to be visible through the colored wall. The total height of the cavity is $h_c = 0.55$ mm. The growth of the size of the fibrous layer adjacent to the cavity bottom slows down, and at $t = 0.5$ ms, its width is $d_m = 4$ mm and its height is $\delta_m = 0.42$ mm.

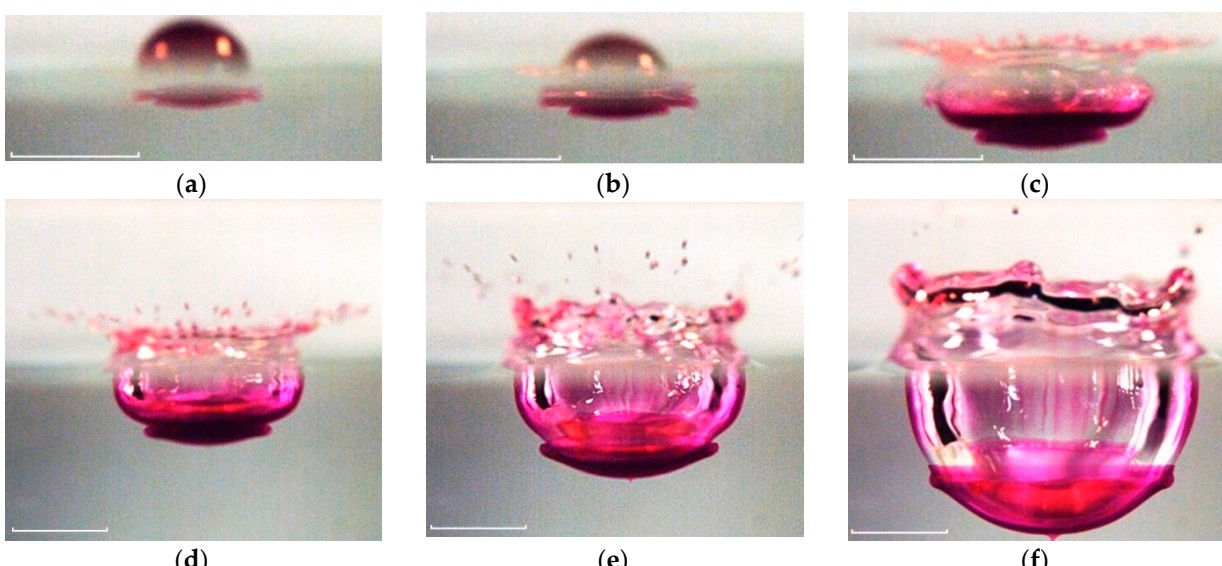

**Figure 9.** Side view of evolution of the impurity distribution pattern on the surface of the cavity and crown when a drop of an aqueous solution of potassium permanganate $KMnO_4$ is immersed in water (the parameters of the experiments are indicated in Figure 7, the tag length is 5 mm). (**a**) $t = 0.25$ ms, (**b**) $t = 0.5$ ms, (**c**) $t = 1.25$ ms, (**d**) $t = 2.25$ ms, (**e**) $t = 4$ ms, (**f**) $t = 8.25$ ms.

All the traditional components of the flow—a cavity, a growing crown with spikes and a spray cloud—are shown in Figure 9c. The diameter and overall height of the cavity with a brightly colored lower layer is $d_c = 8.2$ mm, and the height is $h_c = 1.3$ mm. The dimensions of the adjacent fibrous layer are $d_m = 5.24$ m and $\delta_m = 0.77$ mm at $t = 1.25$ ms

in Figure 9c. The inhomogeneities of the pigment distribution begin to smooth out quickly by the processes of molecular diffusion in the intermediate layer under the cavity bottom ($d_c = 10.3$ mm and the height $h_c = 3.13$ mm in Figure 9d). Here on the side walls of the cavity, the bottom of which is gradually deformed from flat to convex, vertical fibers are viewed.

Further, the rapidly deepening cavity bottom "pushes apart" and throws the remnants of the intermediate layer, the boundaries of which can still be identified at $t = 4$ ms (Figure 9e). The crown begins to fall off, and the capillary waves appear at its edge.

The cavity takes a shape close to spherical at $t = 8.25$ ms (Figure 9f). A system of vertical fibers remains on its side surface. The crown walls are covered with capillary waves. An intermediate layer $\delta_m = 0.3$ mm thick and $h_m = 3.86$ mm high covers the entire cavity bottom with an even layer. The exception is its center, where a thin jet begins to grow.

The three-dimensional structure of the flow in the contact domain of the merging liquids includes thin flat jets running along the cavity bottom to the spikes on the crown edge, and even thinner fibers penetrating the cavity bottom, which form an intermediate layer. At the initial stage, the liquid interface alternately passes along the tops of the fibers with the drop pigment; then, it "jumps" to the tops of the layers of the target liquid, separating the colored fibers. That is, in some domains, it passes along the outer boundary of the fibrous intermediate layer, and in neighboring domains, it passes along the inner one. Qualitatively, we can assume that at the bottom of the growing cavity, the boundary of the coalescence region has a complex three-dimensional piecewise smooth shape.

A sample of shots from the video of the coalescence of a solution of potassium permanganate of higher density, as shown in Figure 10, indicates the scenario stability of the appearance of new components of the flow pattern and illustrates some subtle details of the flow pattern more clearly. The bright color of the spikes with splash droplets flying from their tops confirms the tendency of the rapid flow of the drop pigmented fluid. The velocity of the first splashes in these experiments, which is $u_s = 5.6$ m/s, noticeably exceeds the drop velocity $U = 3.5$ m/s (Figure 10a,b).

The processes of ASPE conversion in a thin layer of merging near-surface layers contact [70,78] provide the inflow of additional energy, which ensures the rapid flow in spikes and sprays. The length of the spikes does not exceed $l_s < 2.7$ mm; their thickness is $\delta_s \sim 0.1$ mm. A colored disk $\Delta r = 0.4$ mm wide between the annular line of coalescence of the drop residue with the target fluid and the growing wall of the cavity and crown is the forming bottom of the cavity with a diameter of $d_c = 5$ mm. The diameter of the crown edge reaches $d_{cr} = 7$ mm at the same time.

The number of spikes decreases with time. Some of them are completely separated from the veil edge. A trough forms in such places (Figure 10c). The heterogeneity of the shape of teeth and spikes indicates the complex three-dimensional nature of fluid flows in the veil. One part of the spikes serves as the continuation of the brightly colored traces of ligaments. The other part is formed as a result of the coalescence of a pair of jets, reaching the thickness of $\delta_j = 0.2$ mm.

Protrusions up to $l_f \sim 1$ mm long are observed at the border of a densely colored domain in the center of the flows. Fast trickles flow out from their tops (Figure 10d). Gradually, as the intermediate layer spreads under the cavity bottom (Figure 10e,f), the background color of the cavity walls becomes more uniform. Triangular elements appear in the flow structure. The heights of the tiers is $l_s = 0.88$, 1.24, 1.67 mm with the thickness of the forming fibers from $0.2 < \delta_s < 0.4$ mm. They are most clearly expressed in the upper part of the cavity in the vicinity of the transition domain where the cavity transfers into the crown, rising above the level of the undisturbed surface (Figure 10e).

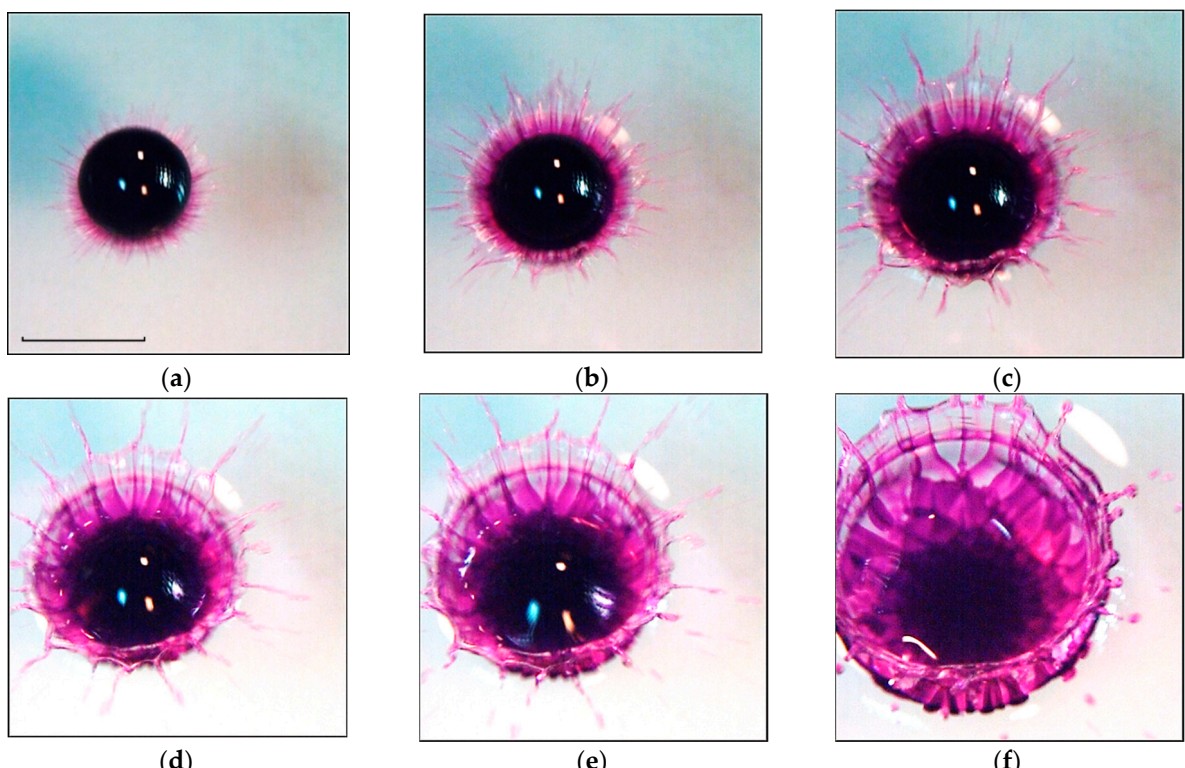

**Figure 10.** Evolution of the flow pattern as the drop of an aqueous solution of potassium permanganate diluted in a ratio of 1:10 merges with water ($D$ = 4.3 mm, $U$ = 3.1 m/s, $E_\sigma$ = 4.2 µJ, $E_k$ =200 µJ, Re = 13,300, Fr = 230, We = 570, Bo = 2.5, Oh = 0.0018, $R_{En}$ = $En_k/En_\sigma$ = 48, $R_W = 1.66 \cdot 10^{-3}$, the tag length in Figure (**a**) is 5 mm, all frames are presented in the same scale). (**a**) $t$ = 0.25 ms, (**b**) $t$ = 0.5 ms, (**c**) $t$ = 0.75 ms, (**d**) $t$ = 1 ms, (**e**) $t$ = 1.25 ms, (**f**) $t$ = 2.5 ms.

As the cavity and crown grow, the number of triangular grid cells on the bottom and walls of the cavern increases, and boundaries appear in their structure, separating ring sets of cells (Figure 10f). The outer border of the cavity loses its smoothness at the bottom of the image. Here, colored sections of fibers begin to be squeezed into the transparent target liquid, the further evolution of which is traced in [88]. When new tiers of the structure are formed, the size of the central spot (a diffusely colored central domain) decreases. Further, as the crown grows, the central spot grows linearly over time until the cavity reaches its maximum depth. When the cavity collapses, the central spot contracts to the center of the cavern remnant and to the top of the ascending reverse jet.

*4.3. Examples of Photographic Registration of the Flow Pattern at the Initial Coalescence Stage of a Free-Falling Drop with the Target Fluid*

The indisputable advantage of video recording is the ability to trace the structural connection and the temporal sequence of individual elements of the flow pattern appearance. The benefit of photo registration within the available technical resources is a higher spatial and temporal resolution. We compared the technical characteristics of two cameras. For the Optronis CR 300 × 2 video camera with a Nikon Nikkor 24–85 mm lens, the distance from the front lens of the device to the center of the shooting area is from 10 to 25 cm, and the exposition is from 1/5000 s. For the Canon EOS 350D camera with a Canon EFS 18–55 mm lens, the distance from the front lens of the device to the center of the shooting area is 5–20 cm, and the exposition 1/4000 s. The spatial and temporal resolution in the video frame (at a frame rate of 4000 fps) and the photo is, respectively, from 30 µm/pix and from 10 µm/pix. Under experimental conditions, photo registration allows analyzing the finer elements of the pattern of a rapidly evolving flow on a moving contact surface that continuously changes its location in space and its shape.

In the experiments carried out, the drops of a solution of potassium permanganate, copper sulfate, iron sulfate, or tap water fall into a cuvette $16 \times 16 \times 7$ cm$^3$ at room temperature. Some physical parameters of solutions are given in Table 1.

**Table 1.** Physical parameters of saturated solutions.

| Solutions | $\rho$, g/cm$^3$ | $\sigma$, g/s$^2$ | $\gamma = \sigma/\rho$, cm$^3$/s$^2$ | $\mu$, g/s·cm | $\kappa_S \times 10^5$ cm$^2$/s |
|---|---|---|---|---|---|
| copper sulfate | 1.206 | 75 | 62.2 | 0.02 | 0.53 |
| iron sulfate | 1.18 | 75 | 63.56 | 0.02 | 0.38 |
| potassium permanganate | 1.04 | 74 | 71.15 | 0.01 | 4.0 |

The registration of the flow pattern is carried out with a Canon EOS350D camera with a 12 mm macro ring, ISO 100, D 5.6, with a minimum exposition 1/4000 s. The distance to the object is 15 cm and the spatial resolution is 13 μm/pix. The inclination of the sight line is $\vartheta = 75°$ to the horizon.

4.3.1. Drop Spreading of Saturated Potassium Permanganate Solution in Water

The main components in the finely structured flow pattern formed during the initial contact of the droplet with the target liquid are a veil or spikes and sequences of small droplets (spray) that fly out both from the top of the spikes and directly from the contact line. The presence of "colored tongues" is explained by the difference in the drop shape from the spherical one at the time of the initial contact. The droplet surface is distorted by Rayleigh oscillations and traveling capillary waves [85], and it is also deformed by a radially spreading air jet [48].

The flow photos at the beginning of the coalescence of a drop of potassium permanganate solution with water are shown in Figure 11a (the inclination of the sight line is $\vartheta = 75°$ to the horizon). In the flow pattern, the "tongues" of the veil are expressed in the range of "10–11" and "13–14 o'clock" with a pronounced outer edge and an inner striped structure. The estimation of the velocity of the longest strokes (traces of droplets flying out from the spikes on the tops of the teeth) is measured by the ratio of their length $l_s \sim 5$ mm to the exposure time $\Delta t = 0.25$ ms is $u_s = l_s/\Delta t \sim 22$ m/s (six times the drop velocity $U = 3.8$ m/s). Careful consideration allows distinguishing thin strokes that have a continuation to the tops of the teeth as well as those in contact with troughs at the boundary of the visible contact domain in the upper part of the drop.

In turn, the upper part of the boundary of the colored fluid is not a smooth line, protrusions and troughs with $\Delta\delta_\varphi \sim 0.2$ mm increments are distinguished in it. The transverse dimensions of the fibers on the veil teeth above the drop are $\delta_f \sim 0.05$ mm or less. The question remains open about the coloring nature of the teeth, which may indicate the formation of a continuous inhomogeneously colored surface of the veil with thin trickles lying on its surface or with separate isolated jets located in the space above and below the veil.

The outer wall of the developed spherical cavity, represented in the lower part of Figure 11b, is covered with protruding fibers $\delta_f \sim 0.5$ mm thick, which are at a distance of $0.2 < \Delta\delta_f < 1.2$ mm. Vortex heads of protruding jets appear on some fibers under the pigmented mesh nodes, which transform into loops as the flow evolves [88]. The original linear structure of the distribution of the drop matter is preserved on the wall of the falling crown remnant, which is covered with short capillary waves and lies in the upper part of the cavity.

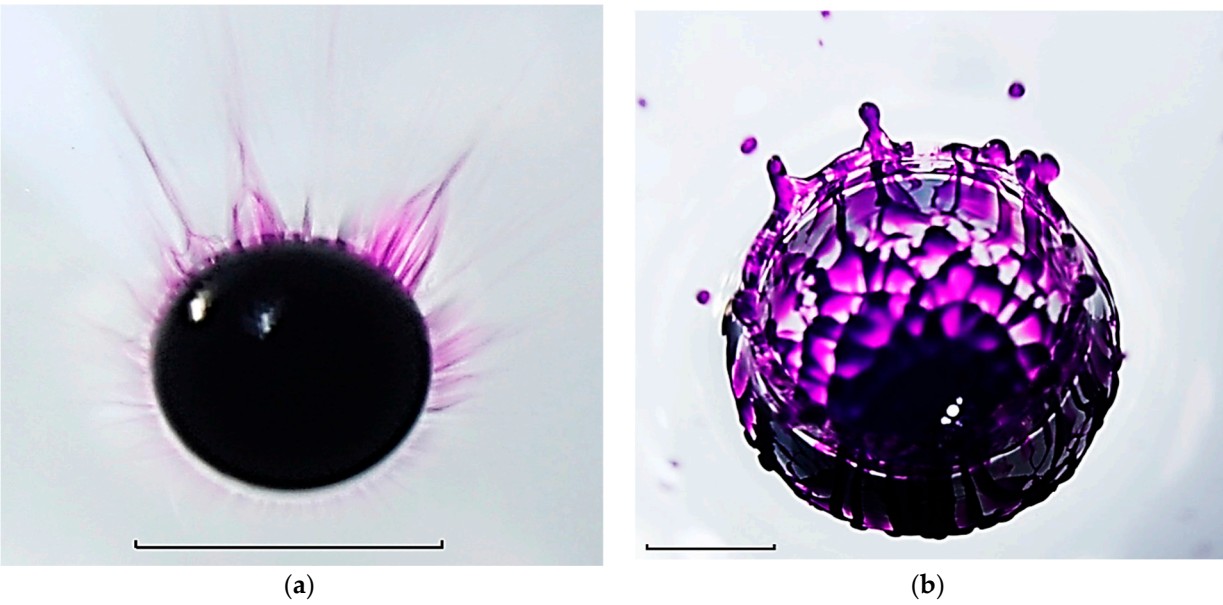

**(a)** **(b)**

**Figure 11.** A drop of saturated aqueous solution of potassium permanganate falls into the water ($D = 4.3$ mm, $U = 3.8$ m/s, $\rho_d = 1.04$ g/cm$^3$, $\sigma_d^a = 74$ g/s$^2$, $\mu = 0.01$ g/cm $\cdot$ s, $En_\sigma = 4.3$ μJ, $En_k = 313$ μJ, $En_k/En_\sigma = 73$, $W_k/W_\sigma = 25 \cdot 10^{-4}$, Re $= 17{,}000$, Fr $= 350$, Bo $= 2.55$, Oh $= 0.0017$, We $= 880$, the tag length is 5 mm). Forms of the flow geometry: (**a**) image of initial contact with annihilation of contacting surfaces, (**b**) developed reticular formation and linear structures on the cavity surface.

At the cavity bottom, fibers with a thickness from $0.1 < \delta_f < 0.4$ mm form a three-tiered mesh with triangular and quadrangular cells of the size from $1.2 < \delta_m < 2.2$ mm. Traces of cells are also visible in the central, more densely colored layer, under which the adjacent intermediate layer is located, which is shown in Figure 9. Dark and light diffuse strips surrounding the cavity in Figure 11b are shadow images of short circular capillary waves [42].

Either a single trickle flows to the tops of the teeth on the upper edge of the crown along its center, or a pair of trickles flow along the outer edge of the tooth, forming a spike, from the top of which sprays fly out.

### 4.3.2. Drop Spreading of Copper Sulphate Solution in Water

A distinctive feature of the flow during the primary contact of a drop of saturated aqueous solution of copper sulfate with water is also the abundance of rapid small sprays, which is presented in Figure 12a (the inclination of the sight line is $\vartheta = 75°$ to the horizon; registration: Canon EOS 350D with a 12 mm macro-ring, ISO 100, D 5.6).

The maximum length of thin strokes (traces of sprays) reaches $l_s = 4$ mm. Taking the stroke width as the diameter of the ejected droplet, and the length as its movement during exposure time, $\Delta t = 0.25$ ms, it is possible to estimate their velocity by the ratio $u_s = l_s/\Delta t \leq 16$ m/s, which significantly exceeds the droplet velocity $U = 3.8$ m/s. The existence of parallel strokes groups is the evidence of the successive ejection of droplets from the top of the same spike on the edge of the cavity veil, the position of which changes as the crown grows. In this experiment, the spray velocity $u_s$ noticeably exceeds the drop velocity $U$; their ratio is $u_s/U = 4.2$. The fact indicates the importance of the influence of the processes of converting ASPE into other forms on the dynamics and structure of droplet flows [70,86].

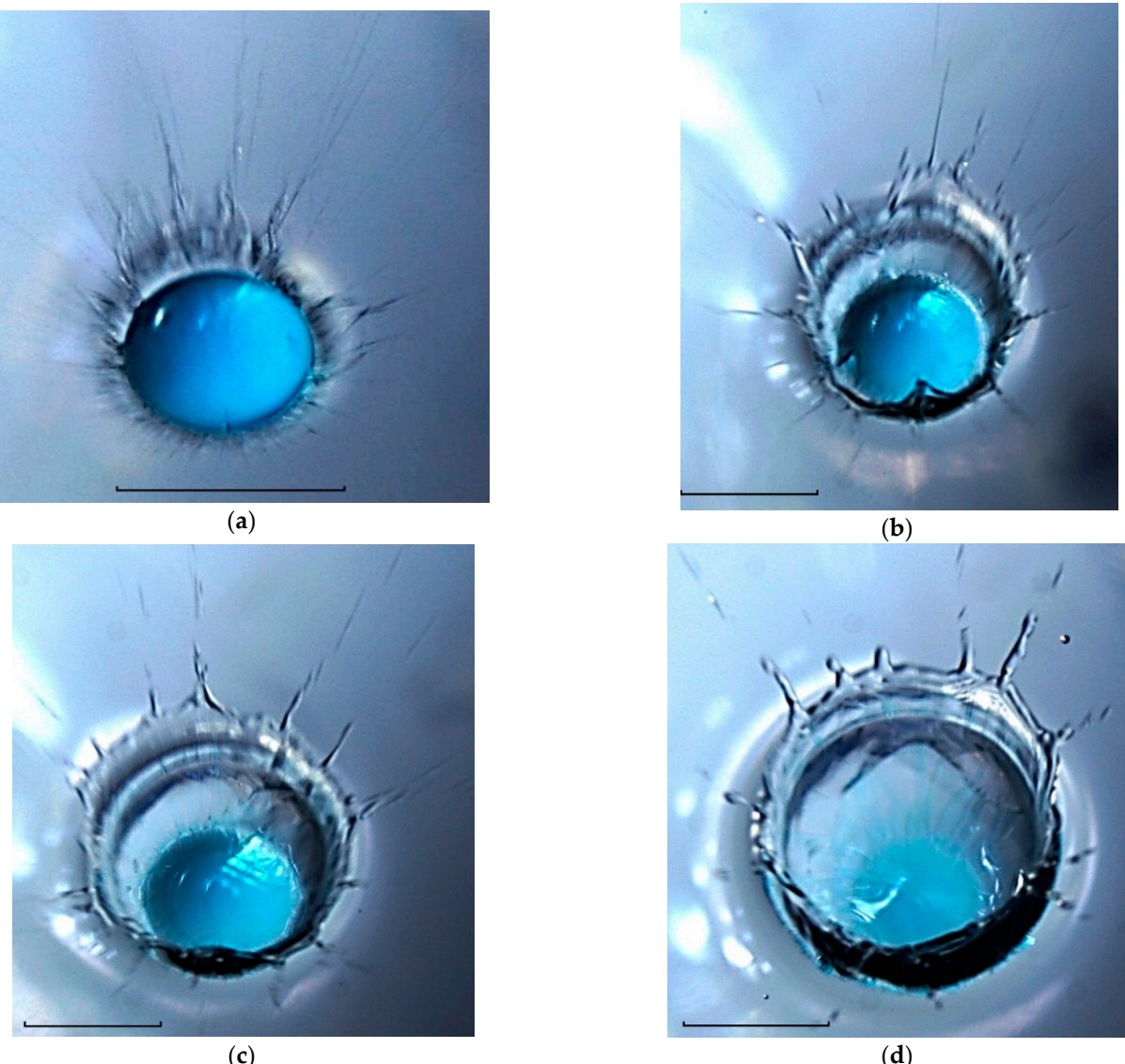

**Figure 12.** Photos of the copper sulfate solution drop spreading pattern in water ($\rho_d = 1.2$ g/cm$^3$, $\sigma_d^a = 75$ g/s$^2$, $\mu = 0.02$ g/cm $\cdot$ s, $D = 4.3$ mm, $U = 3.8$ m/s, $En_\sigma = 4.4$ µJ, $En_k = 363$ µJ, Re = 9900, Fr = 650, We = 1000, Bo = 2.9, Oh = 0.0032, $R_E = En_k/En_\sigma = 83$, $R_W = 3 \cdot 10^{-3}$, the tag length is 5 mm): Typical flow patterns: (**a**)—the image of the first contact with ejecta formation; (**b**)—growing cavity with drop remnant separated by the complex irregular boundary with circular capillary waves, (**c**)—deepening cavity with forming a growing crown, (**d**)—individual protrusions on the drop remnant boundary and linear coloured fibers on the on the surface of the cavity.

Droplet traces (strokes) are located on a pale background of a thin veil $\Delta\delta_v^r = 1.25$ mm wide, adjacent to a darker annular layer $\Delta\delta_{cr}^r = 0.9$ mm wide, in contact with a light strip $\Delta\delta_{ca}^r = 0.14$ mm (the image of the forming walls and cavity bottom). The density of the strokes indicates the high frequency of the process of the sprays generation on the spike tops at the veil or crown edge.

The drop spreads along the bottom as the cavity grows, and in Figure 12b, it is possible to see the boundary of the contact domain of the merging fluids having a complex irregular shape. Basically, the veil and spikes are inclined outward, but in the lower part, there are two sections of the veil, which are inclined inward.

The droplets from their tops fall on the surface of the merging drop and form short capillary waves (in Figure 12b—in sectors for "6" and "8" o'clock [85]). The complex structure of the fluids' contact domain boundary (colored fibers visualizing thin trickles

flowing along the cavity bottom, the crown walls and reaching the spikes on its edge) is preserved throughout the process of droplet coalescence (Figure 12c,d).

Individual protrusions (precursors of the forming vortex loops [88]) are also visible on the outer cavity wall in the lower part of Figure 12b,c.

In the enlarged images of the contact liquids domains shown in Figure 13, fibers adjacent to the contact line and small-scale inhomogeneities of complex shape with a size of $\Delta l_\varphi = 0.22$ mm are visible, forming a moving boundary of the coalescence domain. As in other experiments, the fibers containing the droplet matter go along the bottom and the walls of the cavity, reaching the tops of the spikes on the veil edge [86].

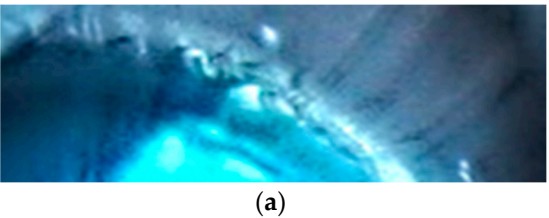 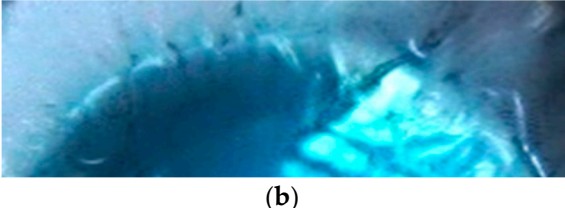

(**a**)                                    (**b**)

**Figure 13.** Enlarged images of the boundary of the droplet coalescence domain; parts of Figure 12b,c. (**a**,**b**) fibers adjacent to the contact line and small-scale inhomogeneities of complex shape.

The colored fibers are traces of fast jets containing droplet matter (ligaments that violate the smoothness of the contact domain boundary of merging liquids). They flow for a long time while maintaining their angular position, leaving connected colored traces on the walls of the cavity and crown.

### 4.3.3. Spreading of an Iron Sulfate Solution Drop in Water

A large number of fast small droplets (sprays) are also formed when a drop of a saturated solution of iron sulfate $FeSO_4$ merges with water. However, the frequency of their ejection here is less than when a drop of copper sulfate solution merges with water. Splashes fly out from the tops of the spikes, to which thin trickles (ligaments) flow, penetrating the domain boundary where the droplet merges with water. Continuous trickles in Figure 14a can be traced at the cavity bottom, the crown walls, above the crown edge.

However, the fine structure of the flow near the boundary of the fluid coalescence domain is more complex here than in Figure 13. In the flow pattern, several groups of teeth are distinguished, the tops of which are wrapped inside the flow domain—two teeth in the left part, two teeth in the right part, and three teeth at the bottom (the angular position is at "3", "6 and 9 o'clock"). Circular disturbances on the surface of the droplet residue in the lower part of the cavity bottom are systems of short capillary waves formed by the coalescence of sprays flying from the spikes on teeth tops [87].

The coalescence boundary (the enlarged image of which is shown in Figure 15) is uneven.

Fine fibers come out of periodic protrusions and troughs between them. The dark circular line on the cavity wall is the crest of a capillary wave running from the coalescence domain boundary.

Fibers (jets, ligaments, trickles) running from the uneven fluids' contact domain boundary were observed in all experiments conducted with colored droplets in the impact coalescence mode. The high (in comparison with the falling drop velocity) speed of the liquid flow and the density of the additional KEn of the flow in the ligaments were provided by the conversion processes of ASPE, which was released during the fluids coalescence and elimination of free surfaces. The transformed energy was stored in the thin volume in which it was concentrated near the free surface.

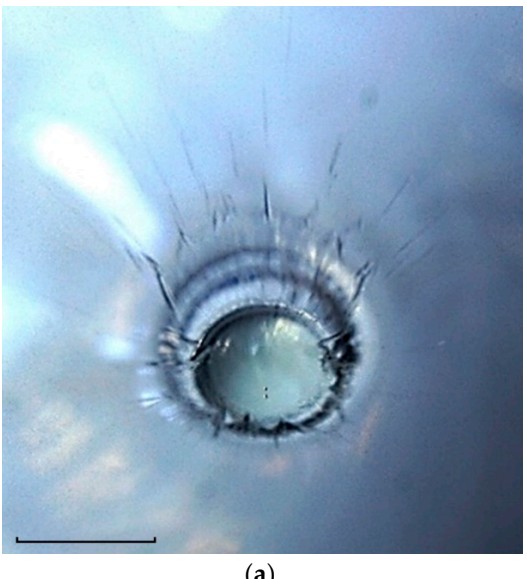 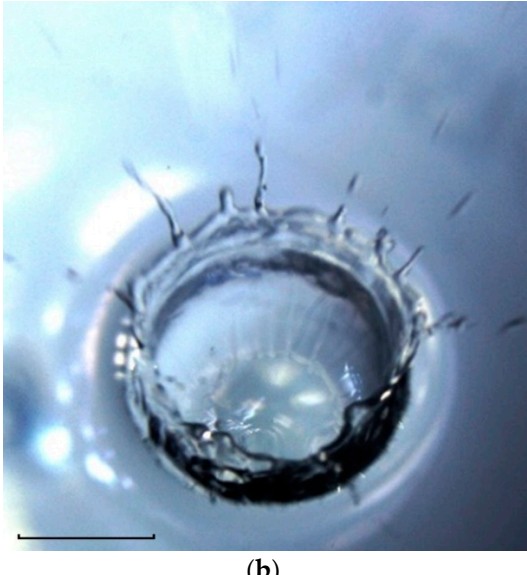

(**a**)                                    (**b**)

**Figure 14.** A drop of an aqueous solution of iron sulfate falls into the water ($\rho_d = 1.18$ g/cm$^3$, $\sigma_d^a = 75$ g/s$^2$, $\mu = 0.02$ g/cm $\cdot$ s, $D = 4.3$ mm, $U = 3.8$ m/s, $En_\sigma = 4.4$ µJ, $En_k = 355$ µJ, $R_E = En_k/En_\sigma = 81$, $W_k/W_\sigma = 2.7 \cdot 10^{-3}$, Re $= 9650$, Fr $= 350$, Bo $= 2.85$, Oh $= 0.0032$, We $= 980$, the tag length is 5 mm): (**a**) the complex form of the boundary of the coalescing fluids; (**b**) fine jets of coloured fluid on the cavity surface.

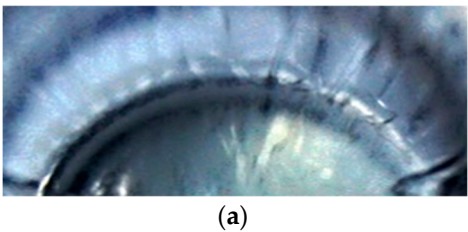 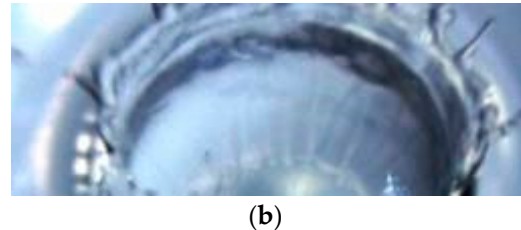

(**a**)                                    (**b**)

**Figure 15.** Enlarged images of the contact domain boundary of a merging saturated iron sulfate solution drop with water (parts of Figure 13). (**a**) fast jets forming liear sructure on the cavity surface; (**b**) traces of jets—ligaments on the bottom of the cavity.

4.3.4. Natural Shadowgraph Patterns of Water Droplet Spreading in Water

Changing the angles of inclination and the position of the water surface causes a redistribution of light flows and leads to the formation of the flow components' distinctive images. Shadowgraph methods, which have been widely used in the study of fluid flows for almost two centuries [90], can be also employed in the study of droplet flows. The resulting large gradients of the refractive index allows tracing the evolution of a flow fine structure under the cavity water droplet in water by the direct shadowgraphy method [84]. The photos presented in Figure 15 show the possibility of visualizing fibers (traces of trickles) on the surface of a complex-shaped liquid not only by tint but also by observations of the redistribution of image illumination.

In the natural shadowgraph flow image shown in Figure 16a, the merging drop residue in the center of the cavity is outlined with a dark line (the coalescence domain boundary). A light strip, which is wall and the growing cavity image, is penetrated by separate jets: ligaments. Thin jets that are generated in the domain where the droplet merges with the target fluid liquid deform the bottom. One group of jets penetrates through the cavity walls, while another part flows along a cavity bottom and is continued on the crown surface, then in the veil, and finally forms dark spikes protruding from the veil edge. They are especially distinguished in the upper right quarter of the flow. Sprays fly out from the spike tops and are registered as thin strokes, the length of which reaches $l_s = 4.25$ mm. The maximum

ratio of splash and drop velocities is $u_s/U = 4.5$. A large number of strokes above the crown and the veil is evidence of a multi-tiered level of sprays distribution flying out from the spikes on the edge of the veil or the crown [86].

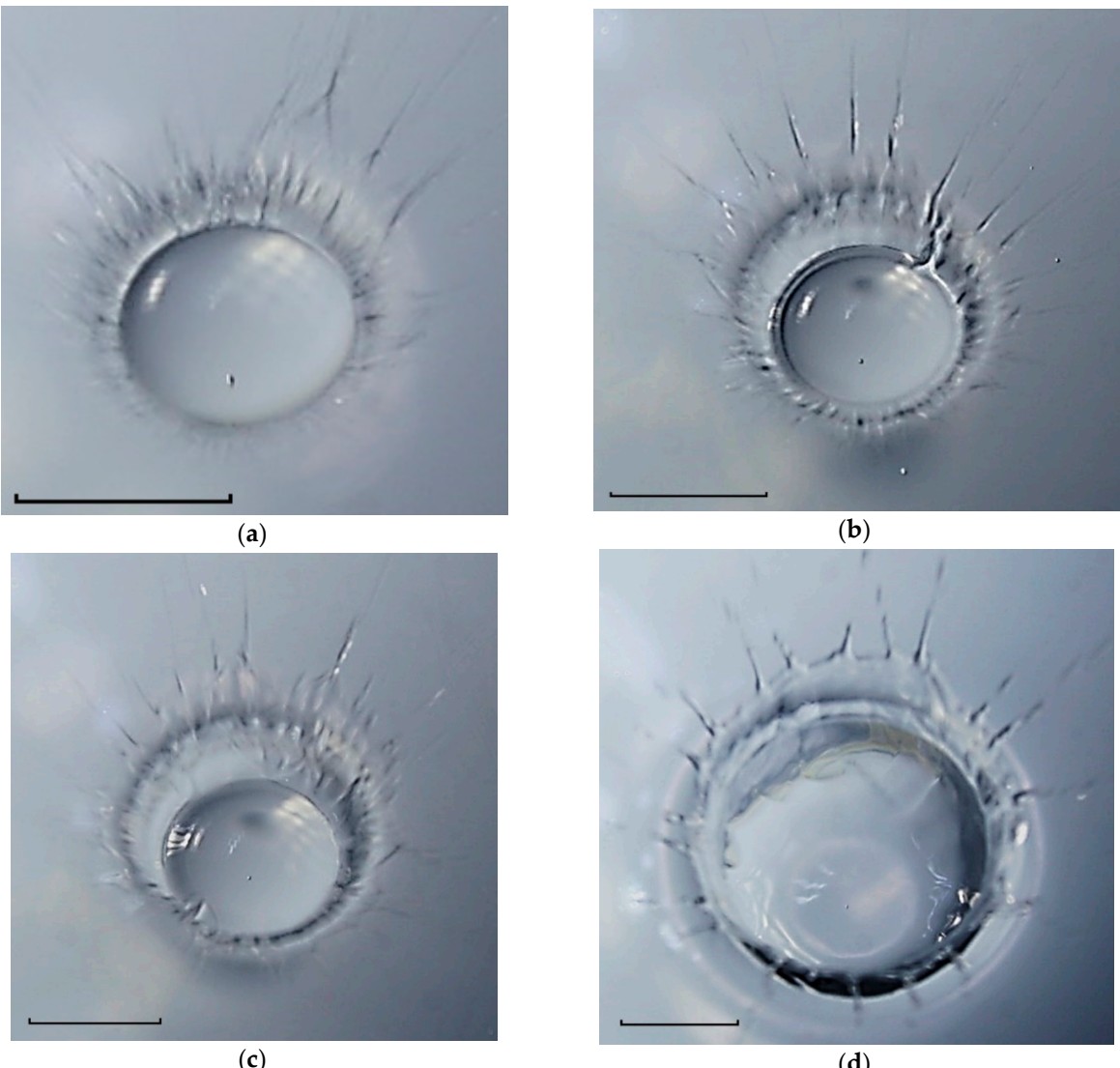

**Figure 16.** Photos of the drop water coalescence with water in the impact mode ($\rho_d = 1$ g/cm$^3$, $\sigma_d^a = 73$ g/c$^2$, $\nu_d = 0.01$ cm$^2$/s, $D = 4.3$ mm, $U = 3.8$ m/s, $En_\sigma = 4.2$ μJ, $En_k = 240$ μJ, Re = 14,600, Fr = 270, We = 690, Bo = 2.5, Oh = 0.0018, R$_E$ = $En_k/En_\sigma$ = 57, R$_W$ = $2 \cdot 10^{-3}$, the tag length is 5 mm): (**a**) fine sprays, spikes, veil, and jets on the contact line of coalescing fluids; (**b**) spikes, veil, set of jets and sharp contact line; (**c**) growing cavity, crown, veil, and spikes on its outer edge; (**d**) deep cavity with drop remnant, crown, and rare spikes.

The contrast of the flow components' images is determined by the inclination of the contacting surfaces, the location of the light sources and a photocamera [86]. An uneven section of the boundary in the range of angles "10–12 o'clock" allows us to consider the visible boundary as the actual edge of the spreading drop. The boundary is radially stretched by thin-layer flows localized in the contact domain of the cavity shell and the near-surface layer of the target fluid [62].

The resulting flows accelerate at the coalescence domain boundary, where the free surfaces are eliminated and the releasing ASPE is transformed into other forms [70]. In this case, together with the matter of the contacting media, a part of the extensive KEn of the merging drop is captured by sprays. It is the rapid release of the ASPE that is the driver of

the flow. At the same time, a noticeable contribution to the fast flow energy is introduced by the energy of the drop mechanical movement.

The spikes on the edge of the cavity can be a continuation of a single trickle (the upper left part of the flow); they can form during the coalescence of two edges of the veil (three spray lines in the upper right part of the figure). The question as to whether the observed structures are continuous jets or a droplets collection consisting of a sequence of small components is still open.

As the drop merges, the number of spikes on the veil border decreases, their thickness increases, and the distribution of illumination in the upper part of the image takes on a diffuse character due to the reconfiguration of a large number of jet elements (Figure 16b). Single jets can be seen more clearly in the lower part of the image. In the upper right corner of the angular position at "2 o'clock", an inverted jet is visible, the top of which falls on the surface of the merging drop. Two clear contours appear in the flow pattern, one of which is the transition from the bottom to the cavity wall, and the second one is the boundary of the spreading drop at the cavity bottom. The images of the fibers in the lower part of the figure emphasize their origin at the contact domain boundary of the droplet residue with the target liquid.

Gradually, the spikes on the upper edge of the veil thicken, and the "strokes" (traces of more and more slowly flying droplets) shorten (Figure 16c). At the same time, the inhomogeneous contrast of dark fibers on the walls of the crown and the cavity is preserved, which is determined by the shape of the surface and lighting conditions. The circular domain boundary where the droplet merges with the deformed liquid surface is visualized as well. In the lower part, the contact of the fibers with the contact line of the merging liquids at the cavity bottom is traced. The sequences of circular lines on the surface of the droplet residue, most pronounced in the sectors at "10" and "14 o'clock", are images of a group of circular capillary waves running to the top of the flow from the expanding boundary of the contact merging fluids domain. A group of capillary waves in an angular position at "7 o'clock" on the surface of the drop residue is a trace of contact of the ejected spikes facing the flow center with the surface of the drop residue [87]. As the flow evolves, the inhomogeneities of the veil structure are smoothed out. The components of the structure on the crown edge and the cavity walls are enlarged.

The formation of additional structural components is accompanied by the emergence of new fine elements in the flow pattern, which, in particular, are expressed as light lines at the cavity bottom in the circular line vicinity of the drop residue coalescence in the "2–5 o'clock" sector. Their geometry, which correlates with the shape of the intermediate layer edge in Figure 4 at $t = 2.75$ ms, confirms the formation of an intermediate layer under the cavity bottom in the impact mode of merging a water drop with water. The layer is created due to the mixing penetrating substance of the drop with the target fluid.

At first glance, the identical physical properties of the contacting liquids in this experiment (tap water) exclude the formation of contrasting images of flow boundaries in the fluid thickness. However, the fact of their reproducible registration in independent experiments [84] indicates the necessity for a more thorough study of the physical nature of the internal boundaries' contrast formation. Perhaps the boundaries reflect large gradients in temperature distributions. Evaporative cooling lowers the surface temperature of a free-falling drop and creates a liquid layer with a changed refractive index, which persists during the flow evolution and is diluted by molecular diffusion processes, as well as the concentration gradient layer in Figures 5 and 8.

A group of light ring strips in the sector of "7–9 o'clock" visualizes capillary waves running along the cavity wall. Radial strokes on the surface of the drop residue are traces of falling sprays. Vertical inhomogeneities of the structure of the cavity side wall in the "10–13 o'clock" sector are the traces of ligaments (thin jets flowing from the fluids coalescence domain boundary to the spikes on the edge of the crown or merged veil, the bases of which thicken over time, and the lengths decrease). Accordingly, the droplet sizes are growing, more and more slowly flying out from their tops.

## 5. Discussion

From the first experiments up to the present, most of the attention has been paid to the study of either the geometry of the free surface [3,91] or the motion of the forming ring vortices and vortex systems [1,2,22,23] when studying the flows generated by a drop falling into a liquid. To visualize the flow, the methods of "bright dot" or "backlight" illumination and "black and white flow pattern registration" are mainly employed [86,89]. During video and photo registration of the black-and-white flow pattern, clear images of the cavity contour were obtained [14], but the distribution features of the droplet matter in the target fluid were not indicated. The results of single experiments on the color registration of the fibrous pattern of the matter distribution of the fallen drop in shallow [63] and in deep fluids [64] have escaped the attention of theorists. Calculations of the matter transfer are traditionally carried out under the assumption of flow axial symmetry, that is, the axial uniformity of the drop matter distribution on the cavity walls [92].

In modern algorithms for calculating droplet flows, the system of fundamental equations with kinematic and dynamic boundary conditions on a free surface [19–21] is replaced by a model system of constitutive equations [46,93]. They accepted assumptions [94] that allow additional expressions simulating the effect of surface tension to be introduced into the equations. Many references (the number to [94], which is the methodological basis for the constitutive description of the liquid surface geometry in droplet flows [46,93], exceeds 2000) illustrate the popularity of the approach. However, the reduction in the system of fundamental equations, in which only the momentum transfer equation is preserved (and at the same time modified), impoverishes the completeness of the flow description, since it does not allow calculations of the energy and matter transfer.

Gibbs' idea of the existence of a diffuse interface layer of contacting miscible liquids of finite thickness is actively used in numerical calculations of droplet spreading over the surface of a solid body. The dependence of internal energy on the "phase measure" and its gradient was introduced in the first mathematical model of a diffuse layer proposed by Van der Waals at the end of the XIX century, [95]. Later, the representation of internal energy was repeatedly used in the development of drop impact flow calculation programs. The technique [95], modified in [96,97], was later actively employed in the formulation of experiments [98], numerical calculations [99,100], as well as in a comparison of the effectiveness of hydrodynamic and kinetic approaches to describing the structure of a diffuse contact layer when spreading droplets on a solid surface [101] and calculating the cavity shape in a liquid [3].

However, a significant difference in experimental conditions does not allow us to transfer the results [95,101] to the analysis of the coalescence drop pattern with the target fluid at rest. When spreading on a solid impermeable surface, the "no-slip conditions", slippage or non-flow conditions that limit the spatial fluid droplet distribution are met. The solid surface remains stationary and does not lose matter. When the droplet spreads, the contact surface of the target fluid actively moves. The ASPE released at the coalescence transfers into other forms including energy of thin flows and outgoing sprays, which contain both contacting fluids.

In all the experiments carried out, the registration of the flow pattern and the shape of the contact domain boundary of merging miscible liquids is implemented in the impact coalescence mode at $En_k > En_\sigma$. Here, in contrast to the intrusive mode [62], the cavity starts to form from the moment of initial contact. At the same time, the droplet disintegrates into separate thin jets. At the initial stage, some of the jets fly out into the air and form sequences of small droplets, the inclination trajectory angle of which gradually increases to the horizon. The velocity modulus of fine splashes is greater than the drop velocity at the time of primary contact.

Some generated jets cross the boundary of the fluid coalescence domain and flow along the deformable surface of the target fluid. The jets reach the edge of the crown or veil and form the teeth on it. In the center of the teeth there are spikes, from the tops of which new portions of sprays fly out. The size of the drops flying out from the tops of

the spikes (the continuations of the jets) grow over time, and the speed decreases rapidly. Some groups of drops flying out from the teeth tilted inward fall on the bottom part of the spreading drop. Since the jet formation sites at the fluids' coalescence domain boundary at the bottom of the growing cavity retain their angular position for a long time, the jets maintain their shape. The remaining colored traces of the ligaments form a linear pattern on the walls of the cavity and crown. At the final stage of the drop spreading, a reticular pattern is formed at the cavity bottom, the size and structure of which depend on the composition of the drop and its shape at the time of initial contact.

Other jets penetrate the bottom of the growing cavity and form a structurally isolated area under its bottom, in which thin fibers containing liquid droplets are separated by layers of the target liquid. Molecular processes of mutual diffusion on the developed contact media surface provide a rapid equalization of the density difference of interpenetrating liquids and form a layer of intermediate density liquid adjacent to the cavity bottom. As the cavity grows, the intermediate layer homogenizes and thins. The rate of density equalization depends on the properties of the medium; for example, the heterogeneity of the distribution of potassium permanganate in it is lost faster than that of alizarin ink. The existence of fine jets at the cavity bottom and the intermediate layer of the cavity is visualized even when a water drop spreads in the water.

The fine structure of the coalescence domain boundary depends on the properties of the fluids and the conditions of the experiment. When saturated solutions of some metal salts (copper and iron sulfate, potassium permanganate) merge with water, the boundary is uneven: there are individual protrusions and troughs, and thin fibers are distinguished in it. The boundary of the ink solution drop coalescence domain is smoother. The complexity of the shape of the boundary is explained by the possibility of forming a large number of multidirectional thin jets, whose flow accelerates the processes of ASPE conversion occurring in a wide range of spatial scales—from the sizes of molecular clusters ($\delta_c \sim 10^{-6}$ cm) to the scales of non-stationary ligaments $\delta_U^\nu \sim 10^{-2}$ cm and more.

## 6. Conclusions

A series of experiments was carried out to visualize the fine structure of the flow in the vicinity of the fluid contact domain boundary when a free falling drop merges with stationary water in a cuvette. The beginning of spreading of a free falling drop of water, dilute ink, solutions of potassium permanganate, copper sulfate, and iron sulfate in deep water was visualized.

In the impact flow mode, when the KEn exceeds the ASPE of the droplet, the cavity begins to form from the moment of initial contact of liquids. In all experiments, the drop loses its continuity and breaks up into thin jets that fly out into the air, flow along the fluid surface or penetrate the cavity bottom.

Small fast droplets fly into the air from the tops of thin jets formed directly on the fluid coalescence domain boundary, or they are ejected from the edges of the spikes, which are located on the tops of the teeth of the veil or crown. The droplet sizes grow over time, and the velocity decreases.

Thin, fast jets flowing along the cavity bottom retain the radial direction for a long time. Their traces form linear and reticular structures at the cavity bottom.

The jets penetrating the cavity bottom form an intermediate layer with their own physical properties. The size and shape of the intermediate layer as well as the degree of uniformity of the matter distribution are changing rapidly.

The jet formation causing the loss of continuity of the cavity bottom and the contact line of merging liquids is associated with the rapid transformation of the available potential surface energy and other components of internal energy into other forms—variations in pressure, temperature, redistribution of matter and acceleration of fluid flows acting together with available KEn.

Further improvement of the experimental methodology in terms of expanding the range of lengths of illuminating electromagnetic waves (toward ultraviolet and X-rays),

as well as increasing the spatial and temporal resolution of the instruments, will clarify the mechanisms of transmission of energy, momentum and matter when droplets merge with a moving liquid and liquid at rest, which play an important role in modern industrial technologies and environmental processes.

**Author Contributions:** Conceptualization, Y.D.C.; methodology, Y.D.C. and A.Y.I.; experimental investigation, A.Y.I.; data curation, Y.D.C.; writing—original draft preparation, Y.D.C. and A.Y.I.; writing—review and editing, Y.D.C.; supervision, Y.D.C.; project administration, Y.D.C.; funding acquisition, Y.D.C. All authors have read and agreed to the published version of the manuscript.

**Funding:** The work was supported by the Russian Science Foundation (project 19-19-00598-P, "Hydrodynamics and energetics of drops and droplet jets: formation, motion, break-up, interaction with the contact surface", https://rscf.ru/en/project/19-19-00598-P/, accessed on 3 August 2023). The experiments were performed at the stands of the Unique Research Facility, Hydrophysical Complex, Ishlinsky Institute for Problems in Mechanics, Russian Academy of Sciences.

**Institutional Review Board Statement:** Not applicable.

**Informed Consent Statement:** Not applicable.

**Data Availability Statement:** Not applicable.

**Conflicts of Interest:** The authors declare no conflict of interest.

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
