# Peer review of "Fine Flow Structure at the Miscible Fluids Contact Domain Boundary in the Impact Mode of Free-Falling Drop Coalescence"

_fluids, doi:10.3390/fluids8100269_

Round 1

Reviewer 1 Report

The comments are attached in this pdf.

Needs improvement on the flow of the manuscript.

Author Response

Dear Mr. Reviewer,

The authors are grateful for your attention to the work, detailed analysis of the content and valuable comments.

On the merits of the comments.

Major comments:

One of my biggest problem with this article is the readability.

“Firstly, the authors explain many, things, which are not very obvious from the represented figures. It is essential that the authors show the phenomenon that they claim to observe in the discussion section.”

Answer.

This is right. Some conclusions are difficult to draw from static pictures; they are better recognized from films. Difficulties in perceiving the text of the manuscript are caused by a number of reasons, including the complexity of the material contained in the color image with new and previously undiscussed features of flow patterns. To improve perception, a Figure 4 with the nomination of structural components has been added.

1“The ratios of the sizes (length and diameter) of individual strokes which are the extensions of spikes show that their velocity noticeably exceeds the drop velocity.” But there was no quantification of the size of strokes or spikes.

Answer.  Corrected.

Added stroke sizes and speed estimates

“With further evolution, the protrusions on the sidewall of the cavity also transform into jets, which form elongated colored loops when the cavity collapses.”

Answer. Corrected.

 Added link to an article containing a detailed description of the process of forming fibrous loops past collapsing cavity.

“The three-dimensional structure of the flow in the contact domain of the merging liquids includes thin flat jets running along the cavity bottom to the spikes on the crown edge, and even thinner fibers penetrating the cavity bottom, which form an intermediate layer.” which is not obviously observable in the figures.

Answer. This is right.

The photographs show only the consequences of the process of formation of thin jets, traces of which form the fine structure of the flow pattern.

“Secondly, it is highly unclear which exact part of the structure the authors are referring to in baby instances. A schematic or representation of the nomenclature they’re using in the description is very much needed for the readers (as it unnecessarily took a lot of back and forth and a bit of literature study to understand them).

For example, the authors mention slope of the cavity or diameter of the crown, veil diameter, jet traces, drop residue diameter, stroke length, central spot diameter. It’ll be much clearer if they point out these in a figure.”

Answer. Corrected.

A Figure 4 has been added with designations of elements and the nomination of characterizing quantities.

“What exactly is the intermediate layer? Is that the small layer of liquid surrounding the cavity?”

Answer. This is right. The layer referred to is a layer of fluid of intermediate density surrounding the cavity. It arises as a result of the introduction of thin fibers with a droplet liquid through the cavity bottom into the target fluid and subsequent diffusion equalization of density.

Line 569: what do the authors mean by vertical fibers on the side of the cavity?

Answer. It is exactly the vertical colored fibers that are visible in the center and at the edges of the cavity.

“Fig.4, seems to have two figures of diameter instead of changes in illumination as mentioned in the caption”.

Thank you for your note and please forgive our inattention. The figure has been replaced.

  1. “The authors have studied the drop spreading of three different liquid drops: potassium permanganate, copper sulphate solution, and iron sulfate solution. They report difference in the characteristic features of coalescence like spray size, thickness of intermediate layer, veil width, etc, but they fail to explain the physics behind such occurrence. Perhaps, maintaining a constant governing non-dimensional number like Ohnesorge different material can ensure obtaining similar characteristics.”

Answer. This is right.

Here only a physical fact is given - the formation of a fibrous structure. The development of a physical model and estimation of structure parameters requires more experiments and theory development.

  1. “The presence of "colored tongues" is explained not only by the conditions of heterogeneity of illumination and registration with an inclined line of vision”.

If inclined line of vision is affecting the visualization, how are the authors accounting for its effect on visualization?”

Answer.

This erroneous explanation has been ruled out.

  1. “Line 595: is the drop velocity mentioned here the average velocity of the drop or the velocity of the moving surface?”

Answer.

That's an interesting point, thanks. Without going into a deep discussion about the unobservability and immeasurability of the “fluid velocity,” we note that the reviewer is right – here the velocity of the moving surface of the drop.

  1. “It is not clear why the authors provide an extensive introduction on thermodynamics of this system if they do not include much of their thermodynamic argument for explaining the observed physics.”

Answer.

The effects of rapid conversion of internal energy are used to explain the appearance in flows of individual components whose velocity exceeds the velocity of the contacting drop, and the very effect of drop disintegration into jets and fibers.

Minor comments:

  1. “References number has to be corrected so that the references follow the chronological order of their appearance in the article. Ref. 4 comes before Ref.1 in the Introduction.”

Answer.

Thanks for your note. Corrected.

  1. “Please describe in line 66”.

Answer.

Corrected.

  1. “What’s the difference between and ?

Answer.

No difference. The reported typo has been corrected.

  1. “Is D the droplet diameter?”

Yes, it is. Specified in the text.

  1. “Show the line of sight in the schematic separately or in Fig.1.”

The lines of sighting and flight path of the drop are shown in Fig. 1.

The authors once again thank you for your attention to the work and constructive comments.

With best regards,

Yuli D.Chashechkin

Andrey Yu. Ilinykh

Reviewer 2 Report

The followed suggestions are for considered.

1. What is the function of photodetector in Fig. 1? why is it necessary? The reason should be described in detail. And the order of the numbers in Fig. 1 is chaotic. It is better to list in clockwise or counter clockwise.

2. How to define the zero time that droplet contact the pool surface? And how can the time be  captured exactly?

3. Only the time is listed with each figure in Fig.8, and it is better to point out the time in other evolution figures.

4. Part 2 describes the parameters and the parameter values are given in many figure titles, however, the analysis on the values is deficiency.

5. Three types of solutions are investigated. The possible reasons that the different solutions show different features should be deeply discussed.

minor editing need to be finished

Author Response

Dear Mr. Reviewer,

The authors are grateful for your attention to the work, detailed analysis of the content and valuable comments.

On the merits of the comments.

The followed suggestions are for considered.

  1. “What is the function of photodetector in Fig. 1? why is it necessary? The reason should be described in detail. And the order of the numbers in Fig. 1 is chaotic. It is better to list in clockwise or counter clockwise.”

Answer. A signal when a light beam intersects in a photodetector with a falling drop triggers video recording of the process with a specified delay. Corrections have been made to the text

  1. How to define the “zero time” that droplet contact the pool surface? And how can the time be captured exactly?

Answer. The time is determined by measuring the distance from the lower edge of the drop to the surface of the liquid on the last frame of the video before touching and estimating the speed of the drop.

  1. Only the time is listed with each figure in Fig.8, and it is better to point out the time in other evolution figures.

Answer. Corrected. Time indicated for each frame.

  1. Part 2 describes the parameters and the parameter values are given in many figure titles, however, the analysis on the values is deficiency.

Answer. You are right, the work is descriptive in nature.

  1. Three types of solutions are investigated. The possible reasons that the different solutions show different features should be deeply discussed.

Answer. You're right. The search for an answer to this question will be included in plans for subsequent research. Here, different solutions illustrate one common property of flows in the impact mode - the disintegration of a drop into fibers, one part of which spreads over the surface of the liquid, the other seeps through the bottom of the cavern into the thickness of the target fluid, and the third throws spikes and sprays into the air.

The authors are grateful for your attention to the work and useful comments.

With best regards,

Yuli D.Chashechkin

Andrey Yu. Ilinykh

Round 2

Reviewer 1 Report

I am happy to note that the reviewers have satisfactorily addressed all of my concerns. Moreover, the additional discussion in the Introduction section is  a good addition. 

Reviewer 2 Report

The questions in the original version have been modified, and the present manuscript has been sufficiently improved to warrant publication in Fluids.